# Estimating the concentration of silver iodide needed to detect unambiguous signatures of glaciogenic cloud seeding

Jing Yang[1,2], Jiaojiao Li[1], Meilian Chen[1], Xiaoqin Jing[1,*], Yan Yin[1], Bart Geerts[3], Zhien Wang[4], Yubao Liu[1], Baojun Chen[2], Shaofeng Hua[2], Hao Hu[1], Xiaobo Dong[5], Ping Tian[6], Qian Chen[1], Yang Gao[2]

[1]Collaborative Innovation Center on Forecast and Evaluation of Meteorological Disasters (CIC-FEMD)/China Meteorological Administration Aerosol-Cloud and Precipitation Key Laboratory, Nanjing University of Information Science & Technology, Nanjing, 210044, China.
[2]China Meteorological Administration Key Laboratory of Cloud-Precipitation Physics and Weather Modification (CPML), Beijing, 100081, China.
[3]Department of Atmospheric Science, University of Wyoming, WY, 82071, USA.
[4]School of Marine and Atmospheric Sciences, Stony Brook University, NY, 11794, USA.
[5]Hebei Provincial Weather Modification Center, Shijiazhuang, 050021, China.
[6]Beijing Weather Modification Center, Beijing, 100089, China.

*Correspondence to*: Xiaoqin Jing (xiaoqin.jing@nuist.edu.cn)

**Abstract.** Detecting an unambiguous radar reflectivity signature is vital to investigate cloud seeding impacts. Radar reflectivity change attributed to seeding depends on both the cloud conditions and on the concentration of silver iodide (AgI) particles. In this study, the reflectivity change induced by glaciogenic seeding using different AgI particle concentrations is investigated under various cloud conditions, using a 1D ice growth model coupled with an AgI nucleation parameterization. In addition, an algorithm is developed to estimate the minimum AgI particle concentration needed for a measurable glaciogenic cloud seeding signature assuming there is sufficient supercooled liquid water. The results show that the 1D model captures the ice growth habit compared to available observations, and yields an unambiguous reflectivity change that is consistent with 3D model simulations and previous observational studies. Simulations indicate that seeding at a temperature of about -15 °C has the highest probability of detecting the radar seeding signature. This finding is consistent with the fact that the seeding temperature was about -15 °C or slightly warmer in most documented unambiguous seeding signature cases. Using the 1D model, 2500 numerical experiments are conducted, and the outputs are used to develop a parameterization to estimate the AgI particle concentration that is needed to detect an unambiguous seeding signature. Application of this parameterization to a real case suggests that seeding between -21 °C and -11 °C can possibly produce unambiguous seeding signatures, and seeding at about -15 °C requires the least AgI particle concentration. Seeding at warmer temperatures in precipitating clouds requires an extremely high AgI amount and supercooled liquid water content. The results shown in this study deepen our understanding of the relationship between the AgI particle concentration and radar seeding signature under different cloud conditions. The parameterization can be used in operational seeding decision making of the optimal amount of AgI dispersed.

# 1 Introduction

Glaciogenic seeding using dry ice or silver iodide (AgI) is the major technique to enhance precipitation in mixed-phase stratiform clouds (Rauber et al., 2019). The scientific principle is that plenty of supercooled liquid water exists in mixed-phase clouds, while the concentration of natural ice nucleating particles is low, which limits the amount of liquid water that can be converted to precipitation naturally (Geerts and Rauber, 2022). Introducing more ice crystals through glaciogenic seeding can enhance the Wegener-Bergeron-Findeisen (WBF) process, which means the ice grows at the expense of liquid water, leading to more precipitating snow crystals (Jing et al., 2015). Investigating the seeding impact on clouds and precipitation has always been vital in this research field (Rasmussen et al., 2018; Rauber et al., 2019; Geerts and Rauber, 2022). However, the seeding signature is often immersed in the large variability of natural precipitation. For decades, scientists have made great efforts to detect unambiguous seeding signatures in field measurements (e.g., French et al., 2018; Rauber et al., 2019; Wang et al., 2021; Zaremba et al. 2024).

The first unambiguous observational evidence of precipitation enhancement by cloud seeding is reported by Hobbs et al., (1981). In their study, dry ice was used to seed a nonprecipitating stratiform cloud. A Ka-band cloud radar showed a clear enhancement of radar reflectivity factor (Ze) after seeding, suggesting the formation of ice plumes. In 1990, Deshler et al. (1990) reported another case with evident precipitation enhancement based on radar measurement. Seeding produced echoes of 3-10 dBZ in nonprecipitating clouds. After that case study, unambiguous seeding signatures in stratiform clouds were rarely documented for more than 20 years, especially for the cases in which AgI was used; one exception is Huggins (2007), who used a scanning Ka-band radar to depict the impact of ground-based AgI seeding. Similarly, Jing et al. (2015; 2016) and Jing and Geerts (2015) showed seeding effects based on X-band radar reflectivity data collected in the 2011-12 AgI Seeding Cloud Impact Investigation (ASCII, Geerts et al. 2013), but those studies analyzed the differences in composite radar reflectivity (SEED − NOSEED), rather than instantaneous reflectivity scans. Recently, in three cases from the 2017 Seeded and Natural Orographic Wintertime Clouds: The Idaho Experiment (SNOWIE; Tessendorf et al. 2019), an unambiguous airborne seeding signature was detected using radar and airborne in-situ measurements. In all the three cases described in French et al. (2018) and Friedrich et al. (2021), the natural precipitation was weak, the ice concentrations in clouds were low ($<1$ L$^{-1}$), and seeding AgI near the cloud tops produced a significant enhancement of ice water content (IWC) and surface precipitation, and increase in reflectivity factor (Ze) of 10-30 dBZ after seeding, detected by a X-band radar. Enhancement of radar reflectivity of 10-30 dBZ induced by AgI seeding was also reported in the last few years from other regions (e.g., Dong et al., 2020; 2021; Yue et al., 2021; Wang et al., 2021), and in some cases parts of the cloud regions were completely glaciated after seeding (Wang et al., 2024). Recently, by seeding supercooled stratus cloud with an uncrewed aerial vehicle, Henneberger et al. (2023) provide new observational evidence of precipitation enhancement at temperatures as high as -5 °C. Unambiguous seeding signature was detected using in-situ and ground-based remote sensing instruments when the background noise is low.

However, despite the successful case studies shown above, most field experiments failed to detect the seeding signature. For example, there were 18 intensive operation periods with airborne seeding in SNOWIE, but a clear seeding signature was observed only in three of them (Geerts and Rauber, 2022). In general, airborne seeding of elevated layers of liquid water is more likely to be radar-detectable than ground-based seeding, because scanning radars cannot see impact very close to the ground, especially over complex terrain (beam blockage). In any event, most field measurements before SNOWIE showed no clear signatures of glaciogenic cloud seeding, though statistical analyses in randomized seeding experiments suggest precipitation enhancement by 3%-17%, and in some cases the precipitation enhancement is up to 57% (e.g., Geerts et al., 2013; Pokharel et al, 2014; Jing et al., 2015, Jing and Geerts, 2015, Jing et al., 2016). In these studies, the evidence in individual radar scans is insufficient to tell whether the seeding is ineffective, or the seeding signature is undetectable (Zaremba et al., 2024). We may conclude that the cloud conditions are not optimal or the AgI concentration is too low for producing a signature greater than the natural variability. The efficiency of cloud seeding depends on the cloud conditions. Previous studies suggest that for optimal seeding conditions, supercooled liquid water must be present in the cloud (Geerts and Rauber, 2022); the cloud depth above -5 °C should be deeper than 400 m for ice growth (Manton et al., 2011); the seeding temperature should be lower than -8 °C (Breed et al., 2014), and the concentrations of natural ice crystals should be low (Jing et al., 2016). However, the thermodynamic and microphysical conditions vary significantly in a cloud and differ substantially among different clouds. It is possible also that seeding is effective, but not radar-detectable, because the natural concentration of snow particles dominates the radar signal (Zaremba et al. 2024). Seeding under different conditions and using different AgI particle concentrations may result in different seeding signatures. It would be helpful in seeding operations if we can quickly estimate which region has the optimal seeding condition, and how much AgI is needed for an unambiguous seeding signature.

The purpose of this study is to develop an algorithm to estimate how much AgI is needed to detect the signature of glaciogenic cloud seeding. The basic idea is that Ze is determined by the ice crystal size and concentration, and the WBF process is the major ice growth mechanism in mixed-phase stratiform clouds. Since there are sophisticated theories and well-developed parameterizations for ice diffusional growth, we can model the Ze induced by cloud seeding. By conducting multiple sensitivity experiments with various cloud conditions, we can develop a parameterization that reveals the relationships between AgI particle concentration and the Ze induced by cloud seeding, and using this parameterization we can estimate the AgI particle concentration needed to produce a Ze greater than the natural variability. The algorithm and the results presented in this study can be used to improve cloud seeding operations efficiency.

The rest of the paper is organized as follows: Section 2 describes the ice nucleation and the ice growth model. The evaluation of the model, discussion of the model results, and the parameterization for AgI particle concentration is developed in Section 3. In Section 4, this parameterization is applied to a real case. The conclusions and discussion are presented in Section 5.

## 2 The ice nucleation and growth model

In this study, we use a one-dimensional (1D) model of ice nucleation and growth to simulate the growth trajectory of ice crystals. Similar models have been used in previous studies to investigate ice microphysics (Chen and Lamb, 1994; Korolev and Field, 2007) and to retrieve ice properties with radar measurements (Zhang et al., 2014). In addition, we implement an AgI nucleation parameterization (Xue et al., 2013a; 2013b) in the model to simulate the ice generation by cloud seeding. Using this model, we can estimate the IWC and Ze produced by cloud seeding in mixed-phase stratiform clouds with relatively weak turbulence.

### 2.1 Parameterization of AgI nucleation

To model the ice nucleation through AgI particles, we follow the parameterization described in Xue et al. (2013a). This parameterization is originally developed based on cloud chamber experiments from DeMott (1995) and Meyers et al. (1995). Four ice nucleation modes are considered. For *deposition nucleation*, which is valid when the saturation ratio with respect to ice greater than 1.04 and the temperature colder than 268.2 K, the fraction of AgI seeded that can nucleate ice is:

$$F_{dep} = a(S_i - 1) + b\left(\frac{273.16-T}{T_0}\right) + c(S_i - 1)^2 + d\left(\frac{273.16-T}{T_0}\right)^2 + e(S_i - 1)^3, \tag{1}$$

where $T_0 = 10$ K, $a = -3.25 \times 10^{-3}$, $b = 5.39 \times 10^{-5}$, $c = 4.35 \times 10^{-2}$, $d = 1.55 \times 10^{-4}$, and $e = -0.07$. $S_i$ is the saturation ratio with respect to ice.

The fraction of AgI that can nucleate ice through *condensation-freezing nucleation*, which applies for temperature colder than 268.66 K, can be calculated using

$$F_{cdf} = a\left(\frac{268.66-T}{T_0}\right)^3 (S_w - 1)^2, \tag{2}$$

where $a = 900$. $S_w$ is the saturation ratio with respect to water.

The third mode is *contact freezing*, which is minor compared to the other nucleation mechanisms:

$$F_{ctf} = F_{scav}[a + b(S_i - 1) + c(S_i - 1)^2 + d(S_i - 1)^3 + e(S_i - 1)^4 + f(S_i - 1)^5 + g(S_i - 1)^6], \tag{3}$$

where $a = 0.0878$, $b = 23.7947$, $c = 52.3167$, $d = 2255.4484$, $e = 568.3257$, $f = 2460.4234$, and $g = 263.1248$. $F_{scav}$ is the fraction of the total AgI particles that are scavenged by cloud droplets (Caro et al., 2004). Contact freezing is valid for $S_i > 1.058$ and $T < 269.2\ K$.

And the last mode is *immersion-freezing*, which is valid for temperature colder than 268.2 K,

$$F_{imf} = aF_{imm}\left(\frac{268.2-T}{T_0}\right)^b, \tag{4}$$

where $a = 0.0274$, $b = 3.3$. $F_{imm}$ is the fraction of AgI immersed in cloud droplets but not nucleated. Based on the equations above, we can calculate the concentration of ice generated through cloud seeding given the AgI particle concentration. For simplicity, this 1D model does not incorporate 3D turbulent dispersion of AgI particles. In each numerical experiment, seeding is performed at a given temperature (height), and seeding takes place only at the beginning of each run. The total AgI particle concentration decreases in every time step (1 s) due to the continuous ice nucleation. The nucleated ice crystals that form at the seeding level are either columnar or plate-like depending on background temperature and remain that way as they descend to the surface.

## 2.2 Growth of ice crystals

In mixed-phase stratiform clouds with relative weak turbulence, the WBF process, where ice grows at the expense of liquid water, is the most important mechanism for ice growth, so if there is sufficient liquid water in the cloud, the diffusional growth contributes the most to ice mass and size. However, it should be noted that the WBF process is not always valid, if the updraft is strong enough, the cloud will be supersaturated with respect to water, leading to simultaneous growth of droplets and ice crystals. Korolev (2007) showed the WBF process only applies in weak downdrafts and weak updrafts, therefore, we emphasize that the model used here only applies for stratiform clouds with weak turbulence. In fact, even weak turbulence may occasionally result in supersaturation with respect to water, and radiative cooling at cloud top reduces stability resulting in weak vertical motions at cloud top that may also enhance supersaturation. These factors are not considered, and could in part contribute to the model uncertainties in the results shown in Section 3. In addition to the WBF process, we also consider the riming process, but the model does not incorporate aggregation or secondary ice production (SIP) mechanisms, such as the rime-splintering process and shattering of freezing drops, which may alter ice particle size distributions and radar reflectivity (Yang et al., 2024). By not including these processes, the model may underestimate the reflectivity in conditions where aggregation is a dominant growth mechanism, or in clouds with abundant supercooled water or high ice crystal concentrations as a result of SIP.

According to Mason (1953), the diffusional growth rate of an ice crystal is:

$$\frac{dm}{dt} = \frac{4\pi C(S_i - 1)f_v}{L_d^2/KR_v T_a^2 + 1/D_v \rho_s}, \tag{5}$$

where $R_v$ is the specific gas constant for vapor, $L_d$ is the latent heat of deposition, $K$ is the thermal conductivity of air, $T_a$ is the ambient temperature, $D_v$ is the diffusion coefficient of vapor, $\rho_s$ is the density of saturated vapor, $S_i$ is the saturation ratio with respect to ice, which depends on the liquid phase and temperature, $f_v$ is the ventilation factor:

$$f_v = \begin{cases} 1 + 0.14X^2, & X < 1 \\ 0.86 + 0.28X, & X \geq 1 \end{cases}, \tag{6}$$

where, $X = S_c^{1/3} R_e^{1/2}$, $S_c$ is the Schmidt number, and $R_e$ is the Reynolds number (Hall and Pruppacher, 1976).

In Eq. 5, the capacitance $C$ reveals the shape of ice crystals, which is temperature and supersaturation depended. There are many laboratory experiments investigated the shapes of ice crystals during diffusional growth (e.g., Fukuta and Takahashi, 1999; Baikey and Hallett, 2009). Typically, plate-like ice crystals (e.g., hexagonal plates, sectored plates, dendrites, etc.) form at temperatures warmer than -4 °C, and at -8 °C – -22 °C; column-like ice crystals (columns, needles, etc.) form at temperatures between -4°C and -8 °C (Fukuta and Takahashi, 1999). At temperatures colder than -22 °C, the ice crystals could be either columns or plates (Baikey and Hallett, 2009). Based on laboratory experiments, Chen and Lamb (1994) developed a parametrization of $C$, for plate-like ice crystals,

$$C = \frac{a\varepsilon}{\sin^{-1}\varepsilon}, \tag{7}$$

$$\varepsilon = \sqrt{1 - c^2/a^2} = \sqrt{1 - \phi^2}, \tag{8}$$

and for the column-like ice crystals, $C$ is given by

$$C = \frac{c\varepsilon}{\ln[(1+\varepsilon)\phi]}, \tag{9}$$

$$\varepsilon = \sqrt{1 - a^2/c^2} = \sqrt{1 - \phi^{-2}}, \tag{10}$$

where $a$ and $c$ are the particle radius along the $a$-axis and $c$-axis. For plate-like ice crystals, the $a$-axis is longer than the $c$-axis, while for column-like particles, the $c$-axis is longer. $\phi = c/a$ is the aspect ratio of ice crystal. The change in the surface volume of the ice crystal is expressed as:

$$dV = \frac{1}{\rho_{\text{dep}}} dm, \tag{11}$$

where $\rho_{\text{dep}}$ is the mass density at the time of deposition. Chen and Lamb (1994) showed that $\rho_{\text{dep}}(g\ cm^{-3})$ can be expressed using:

$$\rho_{\text{dep}} = 0.91\exp\left[-3\frac{\max(\Delta\rho - 0.05, 0)}{\Gamma(T)}\right], \tag{12}$$

$\Delta\rho$ is the excess vapor density. $\Gamma(T)$ is the ice inherent growth ratio, which is parameterized based on the data from multiple laboratory experiments (Harrington et al. 2019):

$$\Gamma(T) = \frac{dc}{\phi da}. \tag{13}$$

The initial ice particle is assumed to be spherical. The volume can be expressed as:

$$V = \frac{4}{3}\pi a^2 c = \frac{4}{3}\pi a^3 \phi, \tag{14}$$

$$d\ln V = 3d\ln a + d\ln \phi. \tag{15}$$

The differential representation of $\phi$ is:

$$d\phi = \frac{1}{a}dc - \frac{c}{a^2}da = \left(\frac{dc}{da} - \phi\right)d\ln a = \phi(\Gamma - 1)d\ln a, \tag{16}$$

$$\frac{d\phi}{\phi} = \frac{dV}{V}\left(\frac{\Gamma-1}{\Gamma+2}\right). \tag{17}$$

Based on the equations above, we can estimate the changes in $\phi$, $a$ and $c$ with time:

$$\phi(t + \Delta t) = \phi(t)\left[\frac{V(t)+dV}{V(t)}\right]^{(\Gamma-1)(\Gamma+2)}, \tag{18}$$

$$a(t + \Delta t) = \left[\frac{3}{4\pi}\frac{V(t)+dV}{\phi(t+\Delta t)}\right]^{1/3}, \tag{19}$$

$$c(t + \Delta t) = \phi(t + \Delta t) \times a(t + \Delta t). \tag{20}$$

Although diffusional growth is the most important mechanism for ice growth in mixed-phase stratiform clouds, accretional growth (riming) is also considered in the model. According to Heymsfield (1982), the mass growth rate through riming can be parameterized as:

$$\frac{dm_R}{dt} = AV_t E(D, d)\text{LWC}, \tag{21}$$

where $A$ is the particle cross-sectional area normal to the fall. $V_t$ is the terminal fall velocity of ice crystals. LWC is the liquid water content estimated by assuming adiabatic clouds. $E(D, d)$ is the collection efficiency between an ice crystal with a diameter of $D$, and a droplet with a diameter of $d$. For simplicity, we assume a droplet diameter of 10 μm, which is a typical size of droplets in continental clouds (Wallace and Hobbs, 2006; Wang et al., 2021).

To implement the impact of turbulence on the growth of ice crystals, we use a Gaussian normal distribution with a zero mean vertical air velocity and standard deviations of 0.75 m s$^{-1}$. A similar method has been used in previous studies (e.g., Korolev et al., 2007; Zhang et al., 2014). The vertical velocity of an ice crystal is the summation of particle terminal fall velocity and vertical air velocity. The ice crystals fall from the seeding level, the temperature below the seeding level is calculated using a lapse rate of 5.5 K km$^{-1}$, and the pressure is calculated using the hydrostatic equation. Sensitivity tests using lapse rates of 5 K km$^{-1}$ and 6 K km$^{-1}$ show minor differences in the modelled results of ice mass.

## 2.3 Terminal velocity of ice crystals

The terminal velocity of an ice crystal depends on its mass, size and habit, as well as the ambient air density. In the
present study, we follow the method developed by Heymsfield and Westbrook (2010) to estimate the terminal velocity
($V_t$) of ice crystals, which has been evaluated against observations (Heymsfield and Westbrook 2010; Zhang et al., 2014):

$$V_t = \frac{\eta R_e}{\rho_{\text{air}} D}, \tag{22}$$

where $\eta$ is the dynamical viscosity and $D$ is the maximum size of the projection of the particle perpendicular to the
direction of fall. According to Böhm (1992), $D = 2a$ for a plate, $D = 2\sqrt{ac}$ for column, and $D = 2r$ for a sphere. $R_e$ is
the Reynolds number:

$$R_e = \frac{\delta_0^2}{4}\left(1 + \frac{4\sqrt{X^*}}{\delta_0^2 \sqrt{C_0}}\right), \tag{23}$$

where $C_0 = 0.35$, $\delta_0 = 8.0$.

$$X^* = \frac{\rho_{\text{air}}}{\eta^2} \frac{8mg}{A_r^k}, \tag{24}$$

where the value of $k$ is 0.5 according to Heymsfield and Westbrook (2010), $A_r$ is the ratio of the projected area $A$ of the
particle to the area of the external circle:

$$A_r = A/[(\pi/4)D^2]. \tag{25}$$

Following Harrington et al. (2013), for plate-like particles:

$$A = \pi a^2 (\rho_i/\rho_{bi})^2, \tag{26}$$

and for columnar particles:

$$A = \pi ac, \tag{27}$$

where $\rho_i$ is the density of ice crystal, $\rho_{bi}$ is the bulk density of solid ice (= 0.91 g cm$^{-3}$). The ice shape (plate-like or
column-like) is assumed not changing with height as it falls.

## 2.4 Calculation of IWC and Ze

In this model, we assumed the initial ice size distribution follows the modified gamma distribution (Mace et al., 1998):

$$N_i(D) = N_X \exp(\alpha)\left(\frac{D}{D_X}\right)\exp\left(-\frac{D\alpha}{D_X}\right), \tag{28}$$

where $D_X = 10$ µm, and $\alpha = 2$. Sensitivity tests using different values for these coefficients will be shown later in Fig. 3. $N_i$
is the ice concentration per unit of length, $N_X$ is the number concentration per unit of length at the functional maximum, and

is determined by the ice nucleation by AgI. This initial size distribution is partitioned to 80 bins with isometric radius intervals, and we loop through all the bins to simulate the growth of ice crystals with different initial sizes. Based on the modelled ice mass and size, we can calculate the IWC and $Z_e$:

$$IWC = \int_0^\infty m(D)N_i(D) \, dD, \tag{29}$$

$$Z_e = N_i \int_0^\infty f(D_m)D_m{}^6 \, dD, \tag{30}$$

where, $D_m$ is the melted ice diameter, $f(D_m)$ is the normalized ice size distribution, $N_i$ is the ice concentration. In the present study, the Ze is calculated assuming the Rayleigh scattering, thus the radar signal is more sensitive to the size of particles. For radars with shorter wavelengths, such as the W-band cloud radar, the signal could be sensitive to the ice concentration. In the SNOWIE experiment, both W-band radar and X-band radars were used. The W-band radar saw clear Ze enhancement near the seeding level, while for the X-band radar, the Ze in the ice plume was weak near the seeding level, and increased downwind towards the surface (Friedrich et al. 2021). We will show that the modelled results (Section 3) is consistent with the observed Ze profile using X-band radar.

In turbulent clouds, the ice crystals may disperse horizontally with time, resulting in a reduced ice concentration. In 1D model, we can estimate the change in $N_i(D)$ by solving the 1D dispersion equation:

$$\frac{\partial N_i}{\partial t} = \nu \frac{\partial^2 N_i}{\partial x^2}, \tag{31}$$

where $\nu$ is the turbulent kinematic diffusion coefficient and depends on the turbulent intensity. In mixed-phase stratiform clouds with relatively simple dynamics, $\nu$ is typically smaller than 10 m$^{-2}$ s$^{-1}$ (Pinsky et al., 2018). In clouds with strong wind shear or embedded convections, $\nu$ could be larger, and the ice growth trajectory could be much more complicated. Therefore, we emphasize again that the model described here only applies to stratiform clouds with relatively weak turbulence. In the present study, the value of $\nu$ is randomly selected between 1 and 10 m$^{-2}$ s$^{-1}$. In clouds with strong turbulence or embedded convection, the microphysics and dynamics are much more complicated (Yang et al., 2016a, 2016b).

## 3 Model results

### 3.1 Examples of ice microphysics from the 1D model

First, we evaluate the growth of a single ice crystal simulated using the 1D model. Figure 1 compares the modelled mass and size of a single ice crystal with the laboratory experiments from Takahashi et al. (1991). The laboratory experiments were conducted at various temperatures under a standard atmospheric pressure. The initial ice crystal is assumed spherical and has a radius of approximately 4 μm. In the model simulation, we use the same thermodynamic conditions and initial size of ice (Eq. 28 is used other than Fig. 1) as the laboratory experiments. It is seen from the figure that the modelled results are fairly consistent with the observations. The mass growth rate has two peaks, at -15 °C and -6 °C, and is the highest at -15 °C. The

ice crystal forms at -15 °C is plate-like, and that forms at -6 °C is column-like (Section 2.2, Eq. 7-10). After 25 minutes, the

mass of ice crystal grows at -15 °C is about an order of magnitude higher than that grows at -20 °C. The sizes at different axes depend on the shape of ice crystals. At -6 °C, the ice crystal is column-like, so the $c$-axis is the largest, while at -15 °C, the $a$-axis is the largest as the ice crystal is plate-like. The ice size can rapidly increase from 4 μm to more than 1 mm within 20 minutes at -15 °C and -6 °C. At temperatures lower than -20 °C, the ice growth habit is less sensitive to the temperature. In general, the uncertainty of modelled ice mass and size is less than 20% compared to the observation. This gives us the

confidence needed to use the model to estimate the ice growth.

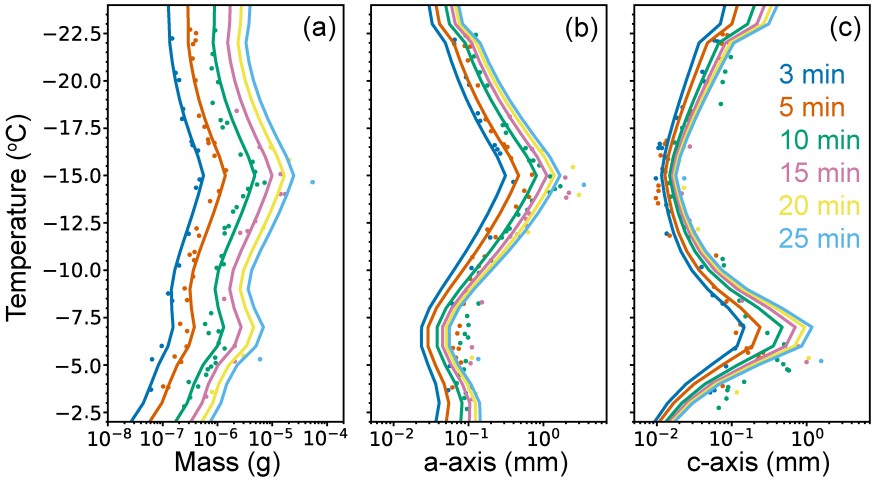

**Figure 1. (a) Growth of the mass of a single ice crystal as a function of time at different temperatures under a standard atmospheric pressure. The dots are from laboratory experiments conducted by Takahashi et al. (1991) and the curves are from model simulations. (b) and (c) are similar to (a) but for the $a$-axis and $c$-axis, respectively.**


According to Figure 1, it is expected that a detectable seeding signature is more probable at -15 °C or colder temperatures. For investigating the seeding signature at a given height, it is necessary to explore how the ice concentration, IWC and Ze change vertically as the ice crystals fall from the seeding level. Figure 2 shows the vertical profiles of ice concentration, IWC and Ze for a seeding temperature of -15 °C, -18 °C, and -20 °C (at a seeding height of 5 km, 5.6 km, and 6 km, respectively). The

results are obtained based on 10 numerical experiments for each seeding temperature. The model runs for 60 minutes in each experiment. The shaded area indicates the 20th-80th percentiles and the solid lines are the mean profiles. The uncertainties are due to the random numbers used in modelling the turbulence effect. Seeding at a relatively low temperature results in a higher ice concentration, and the ice concentration decreases with height. For an AgI particle concentration of 35 cm$^{-3}$, the ice concentrations are 10 times lower. Regardless of the concentration of AgI particles, it is seen that the vertical variations of

IWC and Ze are consistent. Below 4.8 km, IWC and Ze are the greatest when the seeding temperature is -15 °C, and are the lowest when the seeding temperature is -20 °C. Although more ice can be nucleated through AgI at a lower temperature, their

mass growth is much slower than the ice initiated at -15 °C, leading to lower IWC and Ze, which are more sensitive to size than concentration. The vertical variations of IWC and Ze are also related to the terminal fall velocity of ice crystals with different shapes. Ice crystals initiated at -20 °C have a similar scale along the *a*-axis and *c*-axis, thus the ice particle is more

spherical compared to the plate-like ice initiated at -15 °C. The plate-like ice crystals have a small terminal velocity, so they can stay at temperatures near -15 °C for a relatively long time, resulting in a substantial increase in ice mass and size. However, the ice particles initiated at -20 °C have a larger terminal velocity when they fall into the -15 °C level. Based on the results shown in Figures 1 and 2, it is suggested that seeding at a temperature of -15 °C can give a higher probability of detecting the seeding signature using radar. This conclusion is interestingly consistent with the fact that the seeding temperature was about

-15 °C (or slightly warmer in nonprecipitating clouds) in many of the cases with unambiguous seeding signatures detected (e.g., French et al., 2018; Yue et al., 2022; Wang et al., 2024), while at colder or warmer seeding temperatures in precipitating clouds, no clear seeding signature was observed.

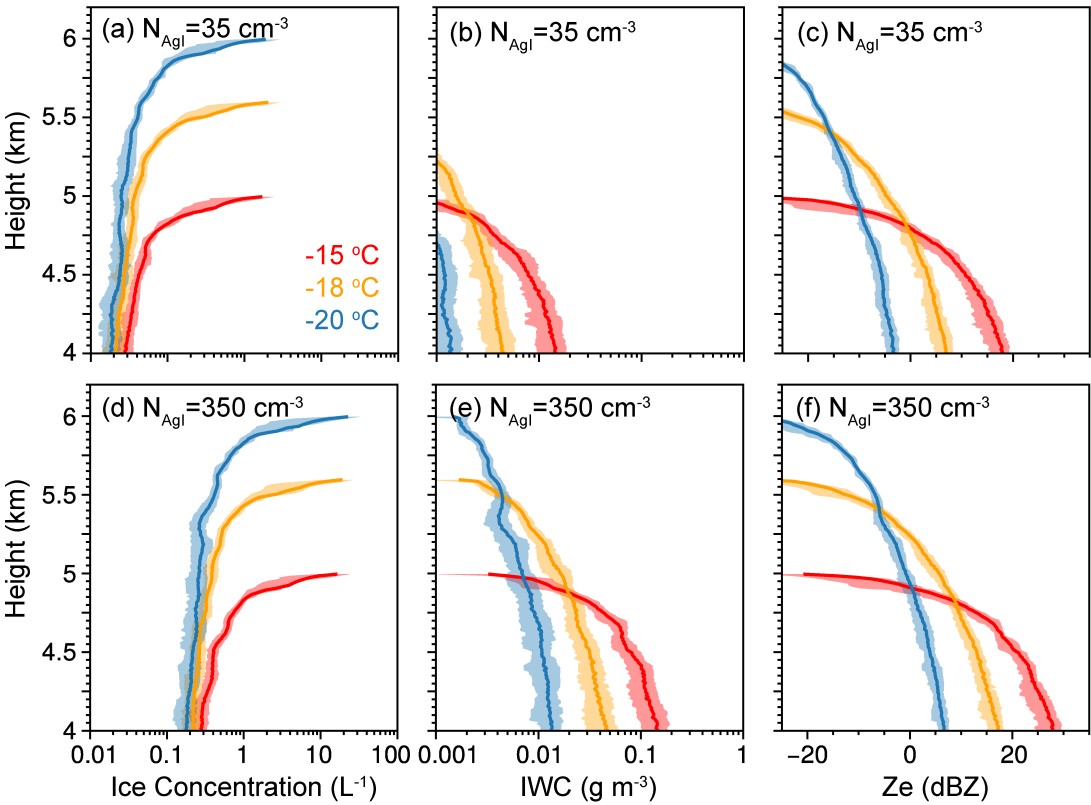

**Figure 2. Vertical profiles of (a, d) ice concentration, (b, e) IWC, and (c, f) Ze from the simulations with different**

**seeding temperatures and an AgI particle concentration of (a-c) 35 cm⁻³, and (d-f) 350 cm⁻³. The results are obtained based on 10 numerical experiments for each seeding temperature. The shaded area captures the 20th-80th percentile range, and the solid lines are the mean profiles.**

In the model, we assume there is sufficient supercooled liquid water and a continuous water supply. However, in real cloud this is not always true, and there is often an upper limit of LWC in clouds. This would certainly affect the ice growth and the Ze profiles. We made several sensitivity tests with a seeding temperature of -15 °C, including different upper limits of LWC (assuming no continuous liquid water supply, Fig. 3a), different time durations for ice growth (Fig. 3b), and different AgI particle concentrations for a limited LWC (Fig. 3c). It is seen from the figure that for a model time of 90 minutes, the Ze decreases with decreasing LWC (Fig. 3a). For a given LWC of 0.2 g m$^{-3}$, ice nucleation and growth in a longer time would consume more liquid water, leading to lower Ze (Fig. 3b), which means the ice formed later on has no sufficient liquid water and vapor to grow. For a given LWC and temporal duration, more AgI concentration does not always produce a larger Ze (Fig. 3c), ice crystals may compete for the limited liquid water and suppress the ice crystal size. In addition, we made sensitivity tests of different turbulent dispersion coefficients and different initial ice particle size distributions (Fig. 3d-f), these are also sources of uncertainties in the model, though the Ze profile is less sensitive to them compared to LWC. These results provide us a better understanding of how the modelled ice mass and Ze may vary due to the different environmental conditions.

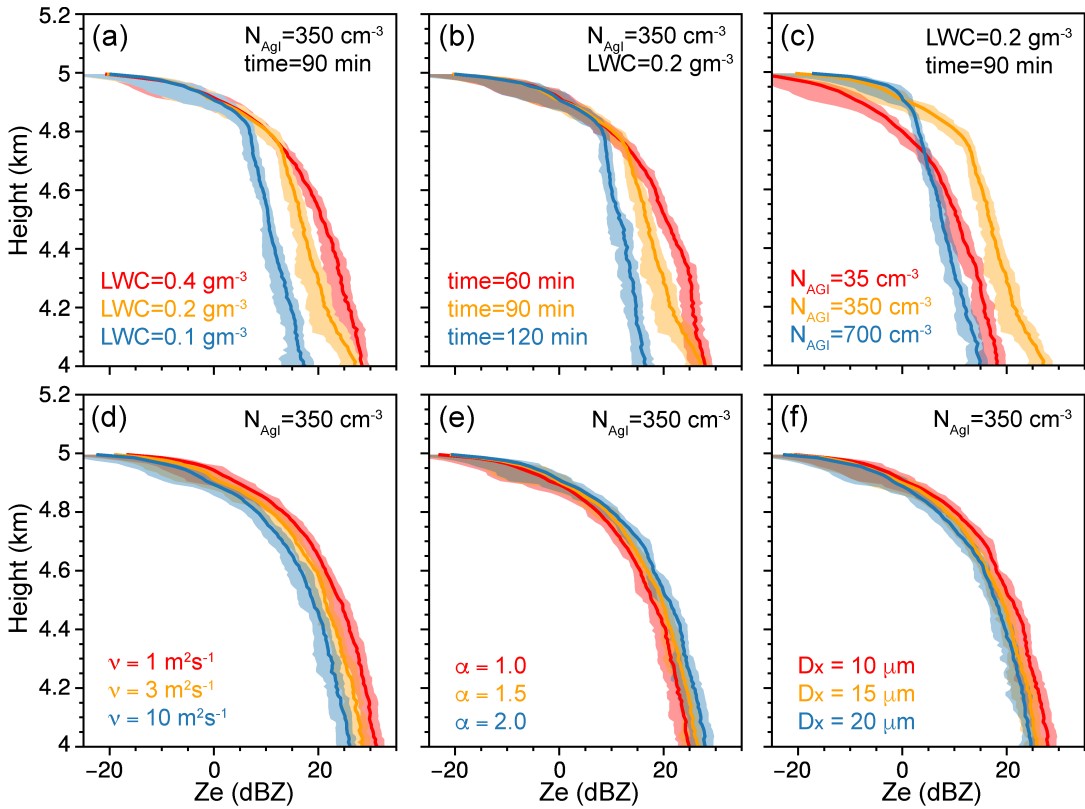

**Figure 3. Vertical profiles of Ze simulation using different (a) upper limits of LWC, (b) temporal durations for ice growth, (c) AgI particle concentrations with limited LWC, (d) turbulence kinematic diffusion coefficients, and (e, d) coefficients in the initial ice particle size distributions. Seeding is perfomed at -15 °C in all the experiments.**

## 3.2 Comparison to 3D model simulation

To further evaluate the 1D ice growth model, it is compared with the 3D large eddy simulation (LES) of stratiform clouds observed on 1 January 2022 in Northern China. The cloud was originally shallow, with a depth of approximately 600 m. The cloud top temperature was approximately -16 °C (Wang et al., 2024). This cloud was seeded near -15 °C, and evident Ze
enhancement was detected by radar after seeding (Wang et al., 2024). The cloud was almost all liquid before seeding, the seeded area quickly glaciated within an hour. More details of measurements can be found in Wang et al. (2024). In this study, we use the LES mode in WRF with periodic boundaries to simulate the cloud. The solid lines in Figure 3 shows the initial profiles of temperature, potential temperature, vapor mixing ratio, and wind velocity above the ground level. Adiabatic liquid water content (LWC) is assumed in the cloud, and the wind shear in the cloud layer (dark-shaded area in Fig. 3) is relatively
weak. To evaluate the performance of the 1D model in simulating deeper clouds, we modified the temperature and vapor mixing ratio data (dashed lines) to allow a deep saturated layer to form. The modelled cloud has a base height of 1.3 km and a top height of 2.9 km (shaded area in Fig. 3). The initial LWC increases from 0 g m$^{-3}$ to 0.2 g m$^{-3}$ from cloud base to top, and rapidly decreases to 0 g m$^{-3}$ above 2.9 km.

In the experiment for the shallow cloud, the model has a horizontal resolution of 50 m, and the domain size is 10 km × 10 km × 4 km. In the deep cloud experiment, the same horizontal resolution and size is used, but in the vertical dimension, we extend the depth to 5 km. In both experiments, 180 vertical levels are used to resolve the vertical structure of the cloud. A ±0.1 K temperature perturbation is added at the cloud base to kick off turbulence. A spin-up time of 30 min is used (Yang et al., 2024), and the ice process is turned off during the spin-up time. Seeding is conducted at 30 minutes along a straight line (with a length
of 10 km in the longitudinal direction) at 1.8 km (-15 °C) for the shallow case, and at 2.6 km (-21 °C) for the deep case. We tested two different AgI particle concentrations (35 cm$^{-3}$ and 350 cm$^{-3}$) in the simulations. The physics schemes used in the simulation include the Revised MM5 surface layer scheme (Jiménez et al., 2012), the Noah Land Surface Model (Tewari et al., 2004), the Rapid Radiative Transfer Model (Mlawer et al., 1997), and the fast spectral bin microphysics scheme (Khain et al., 2004), which can explicitly simulate more complicated microphysics than the 1D model. Cumulus and boundary layer
parameterizations are turned off in the LES. The cloud condensation nuclei (CCN) concentration is expressed by $N_{CCN} = N_0 S_w{}^k$, $N_0$ refers to the CCN concentration at a supersaturation level of 1%, $S_w$ represents the supersaturation with respect to water (%), $k$ is the slope of the CCN size distribution. For continental China, Qu et al. (2017) suggested $N_0 = 4000$ and $k = 0.9$. The warm rain process is not possible with such a high aerosol concentration and weak wind shear in the cloud, since collision-coalescence cannot be kickstarted. Since the observation suggests the cloud is all liquid before seeding, the natural ice
nucleation process is turned off in the model. Thus, ice crystals are only generated due to cloud seeding. Similar to the 1D model, the AgI nucleation parameterization is from Xue et al. (2013a).

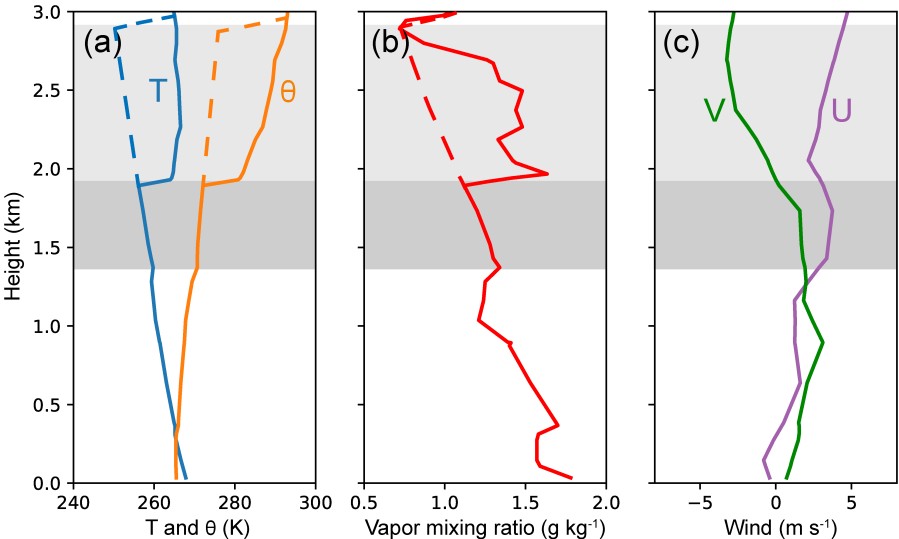

**Figure 4. The initial profiles of (a) temperature and potential temperature, (b) vapor mixing ratio, and (c) U and V components of the wind field. The solid lines indicate the original sounding data used for the shallow cloud, and the dashed lines indicate the modified data used for the deep case. The dark-shaded area (1.3km – 1.9 km) indicates the shallow cloud layer and the light-shaded area (1.3 km - 2.9 km) indicates the deep cloud layer.**

Figure 5a-d shows the Ze, cloud water and ice mixing ratio 30 minutes after seeding in the 3D LES model for the shallow case. Ice crystals, which are initially nucleated near the cloud top, can rapidly grow and fall out of the cloud within 30 minutes. With a lower AgI concentration (35 cm$^{-3}$), the ice mixing ratio is lower than 0.06 g kg$^{-1}$ after 30 minutes, while with a higher AgI particle concentration (350 cm$^{-3}$), the modeled ice mixing ratio exceeds 0.15 g kg$^{-1}$. Since the cloud is all liquid in areas unaffected by seeding, the seeding signature can be well identified once the ice plume forms, either using in-situ measurements or remote sensing measurements (Wang et al., 2024; French et al., 2018). A higher ice mixing ratio implies a higher radar reflectivity. As shown in Figure 5a, with a lower AgI particle concentration (35 cm$^{-3}$), the maximum Ze near the cloud base 30 minutes after seeding is approximately 10 dBZ, while for a higher AgI particle concentration (350 cm$^{-3}$), the Ze exceeds 20 dBZ. The Ze is relatively small near the cloud top and increases from cloud top to base. Below the cloud base, Ze decreases due to ice sublimation. The magnitude of Ze in the core of seeding plumes (5-30 dBZ) is fairly consistent with observational studies in which cloud seeding is operated at -15 °C (Wang et al., 2021; French et al., 2018), indicating the LES model can capture the characteristics of ice growth habit. The ice concentration in such a shallow cloud with top temperatures as warm as -15 °C is typically low (Zhang et al., 2014), resulting in a weak Ze in the natural cloud. The Ze attributed to cloud seeding (Fig. 5) is sufficiently large to be detected by operational weather radars.

For the deeper case, it is seen from Fig. 5e-f that the modelled Ze and IWC are much lower than that in the shallow case, because seeding is performed at a temperature of -21 °C. We show the results 40 minutes after seeding because the ice crystals

need more time to reach the cloud base than that in the shallow case. In generally, the Ze and IWC increases from the seeding level towards the cloud base, the maximum Ze is only -10 dBZ and 1 dBZ for a AgI particle concentration of 35 cm$^{-3}$ and 350 cm$^{-3}$, such low values are often smaller than the signals of natural precipitation. This conclusion is consistent with Fig. 2, therefore, both the 1D model and 3D model suggest a high ice growth rate at -15 °C, and a lower ice growth rate at colder temperatures.

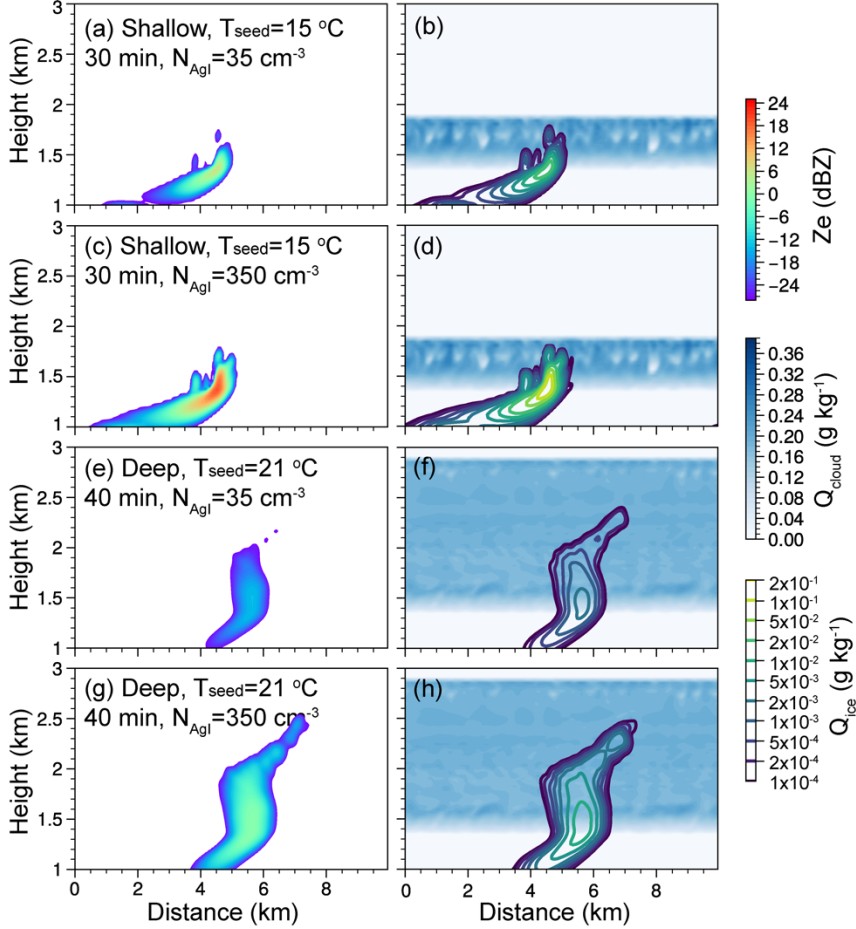


**Figure 5. Cross sections of (a) Ze (rainbow shading), (b) liquid water mixing ratio (blue shading) and ice mixing ratio (contoured) obtained from the 3D LES simulation for the shallow case using an AgI particle concentration of 35 cm$^{-3}$, seeding is performed at 1.8 km (-15 °C). (c) and (d) are similar to (a) and (b) but for an AgI particle concentration of 350 cm$^{-3}$. (e-h) are similar to (a-d) but for the deep case with a seeding level at 2.6 km (-21 °C).**


Figure 6 compares the radar reflectivity from the 1D model and 3D model simulations. The left and right boundaries of the blue shaded area indicate the 95$^{th}$ percentile and the maximum Ze respectively from the 3D model. As shown in the figure, the 1D model is generally consistent with the 3D model, and the oversimplified dynamics do not change the magnitude and vertical

variation of Ze. Near the cloud base, an AgI concentration of 35 cm$^{-3}$ leads to a maximum reflectivity of 11 dBZ (Fig. 6a), and
the Ze increases to 21 dBZ as the AgI concentration increases to 350 cm$^{-3}$ (Fig. 6b). This magnitude is similar for the 1D and
3D model. Therefore, the 1D model is valid for investigating the vertical variation of Ze below the seeding level. Although the
1D model is consistent with the 3D model for the cases presented here, it should be noted that uncertainties are inevitably
present in modelling Ze in both the models. Improving the ice nucleation and growth parameterizations in model is vital for
the purpose of this study. Previous observational studies showed the ice nucleation efficiency of AgI particles vary in different
experiments (Marcolli et al., 2016), and recently, Ramelli et al. (2024) showed the ice growth rates have large variabilities in
seeded clouds using in-situ and remote sensing measurements, indicating that in real clouds the dynamics and microphysics
are complicated, such observational datasets are useful to evaluate and improve ice growth models (Omanovic et al., 2024).

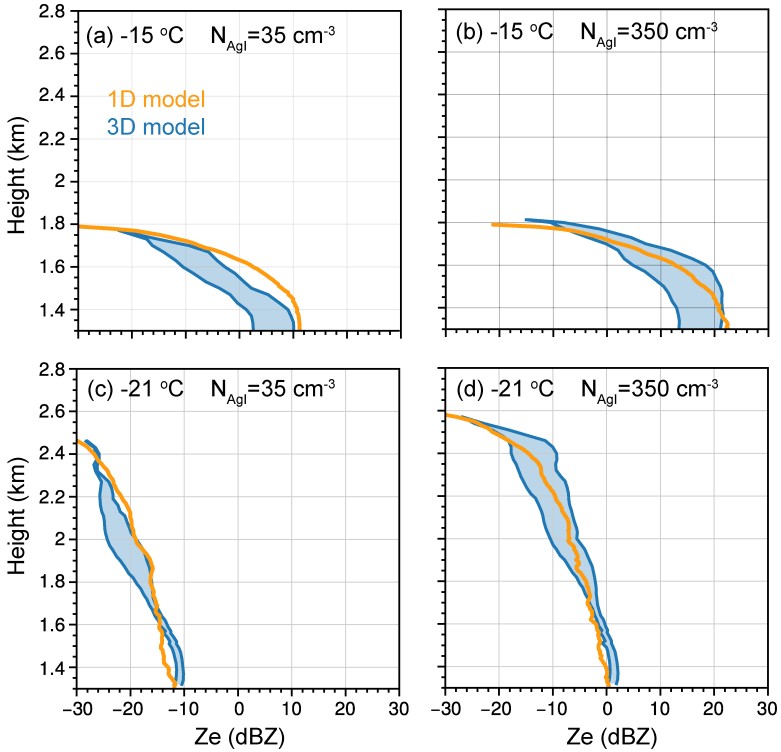

**Figure 6. The vertical profiles of Ze simulated using the 1D and 3D models with an AgI particle concentration of (a) 35**
**cm$^{-3}$, and (b) 350 cm$^{-3}$, respectively. The left and right boundaries of the blue shaded area indicate the 95[th] percentile**
**and the maximum Ze from the 3D model, respectively.**

### 3.3 Parameterizing the AgI concentration needed to detect unambiguous seeding signatures

Since the 1D model can simulate the growth habit of ice crystals reasonably well, we conducted 2500 numerical experiments
with a variety of different seeding temperatures (-30 °C – -7 °C), seeding pressures (350 hPa – 850 hPa), AgI particle

concentrations at seeding level (1 cm$^{-3}$ – 10$^6$ cm$^{-3}$), and depths from seeding level (i.e., the vertical distances between the seeding levels and target levels where we want to detect unambiguous seeding signatures, 500 m – 4000 m). We include very high AgI particle concentration at seeding level in the experiments because it is helpful to have a large parameter space to constrain the parameterization. In addition, in many observational studies, normal AgI particle concentration (10–1000 cm$^{-3}$)

is insufficient to detect the seeding signature, though AgI particle concentration with a magnitude of 10$^5$ – 10$^6$ cm$^{-3}$ is probably not possible in reality. Again, it should be noted that we assume there is sufficient supercooled liquid water in the cloud. Figure 7 shows the modeled Ze (colored) at the target levels as a function of different seeding temperatures, AgI particle concentrations, and depth from the seeding level. Statistically, we can see the temperature and AgI particle concentration both have important impacts on the seeding impacts. The modeled Ze is relatively high at -15 °C with a higher AgI particle

concentration. Clouds with cold top temperatures can be either shallow or deep, for a given seeding temperature, the modelled Ze increases with the increasing depth (similar to the Ze profiles in Fig. 2). Cloud top pressure also influences the growth rate of ice crystals but its impact is minor compared to the other factors.

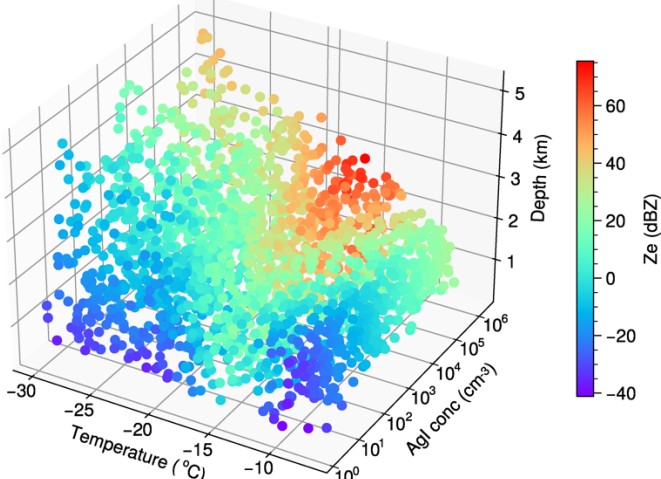

**Figure 7. The modeled Ze (colored dots) at target levels for different seeding temperatures, AgI concentrations, and**
**cloud depth from the seeding level based on the data from 1000 numerical experiments using the 1D model.**

Using the results from the 2500 numerical experiments, we parameterize the AgI particle concentration that is needed to detect the seeding signature. We train the data using a polynomial regression based on 2000 experiments randomly selected from all simulations, then test the parametrization using the remaining 500 experiments. The inputs are seeding-level temperature

($T_{seed}$), pressure ($P_{seed}$), cloud depth from the seeding level ($D_{seed}$), and Ze attributed to seeding at the target level ($Z_{e\_target}$). The output is AgI particle concentration ($N_{AgI}$) at the seeding level.

$$N_{AgI} = f(T_{seed}, P_{seed}, D_{seed}, Z_{e\_target}) \tag{32}$$

Thus, for a given cloud, we can decide how much AgI is needed to seed at different heights to produce a radar signal exceeding the natural variability at the target level. Figure 8a compares the modeled and parameterized AgI particle concentration using the training data. Generally, the data points are along the 1:1 line, and it works well for the test data (Fig. 8b), indicating that the parameterization can reasonably reveal the relationships between AgI particle concentration and Ze for different cloud conditions. In the polynomial regression used for Fig. 8a, 6 degrees (i.e., the maximum power for the input variables) are used. We also tried different values of degree in the regression, as shown in Fig. 8c, a degree of 6 gives the smallest root-mean-square-error (RMSE) and the highest correlation coefficient between the modeled and parameterized AgI particle concentration. The correlation coefficient increases as the degree increases from 1 to 6. For degrees larger than 6, the polynomial regression becomes overfitted.

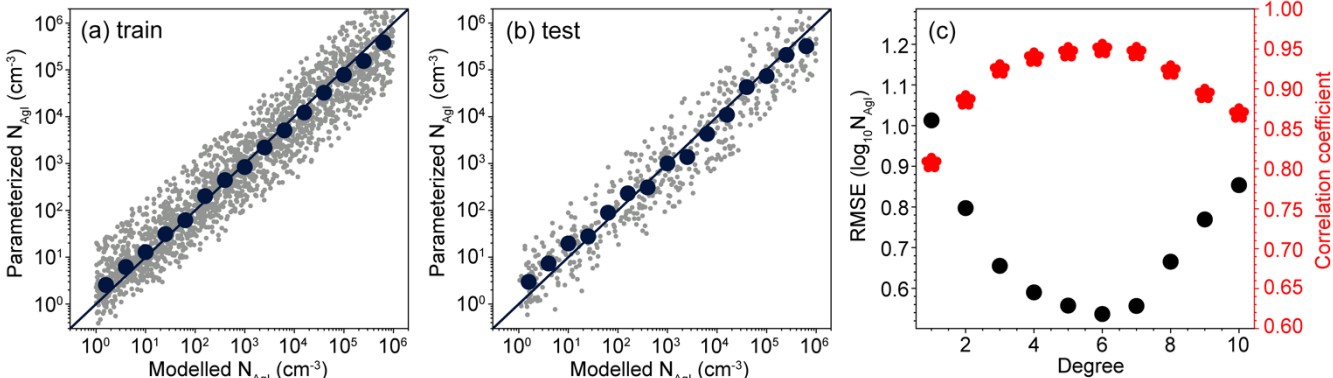

**Figure 8. (a) Scatter plot of the parameterized and modelled AgI particle concentration. The grey dots are the results from the 2000 numerical experiments for training, and the dark blue dots are binned averages. The parameterization is developed using the polynomial regression with a degree of 6. (b) Similar to (a) but for the 500 experiments used for the test. (c) RMSE and correlation coefficient between the parameterized and modelled AgI concentration for different polynomial degrees used in the regression of training data.**

## 4 Application to a real case

### 4.1 Radar observations

In this section, the parameterization is applied to a mixed-phase stratiform cloud with moderate natural precipitation (0.2-0.3 mm h[-1]). The purpose is to estimate the concentration of AgI that is needed to detect the seeding signature in this case. Figure 9 shows the temperature profile and the Ze measured by a Ka-band cloud radar. The cloud was observed in the Hulun Buir region in the northeast of China on 3 August 2023. It formed during the passage of a warm front (near z=1.7 km in Fig. 9a). The cloud was stratiform and deep, with a top temperature of about -25 °C. The freezing level was at about 4.2 km. It is seen that there are large natural variabilities of the observed Ze. Above the freezing level, the largest Ze is about 15 dBZ, and the average Ze is 4 dBZ.

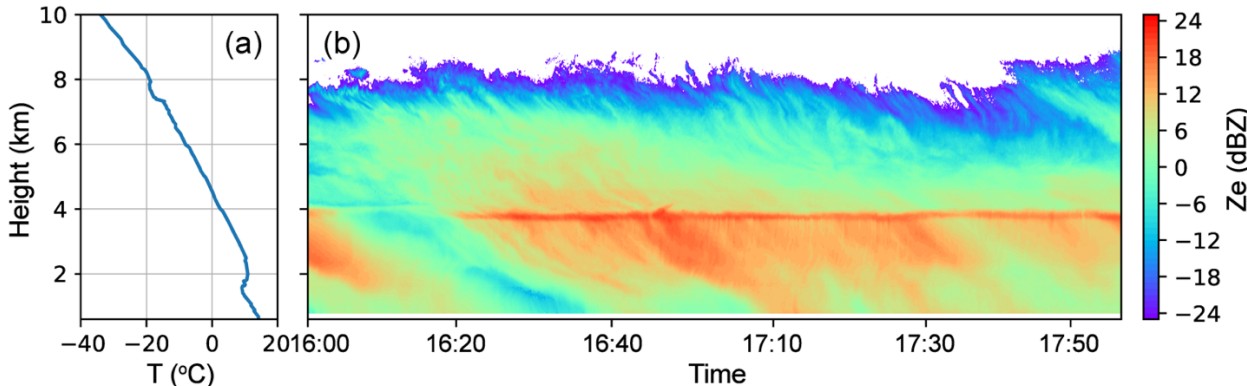

**Figure 9. (a) Temperature profile measured by radiosonde and (b) Ze measured by a Ka-band cloud radar for a stratiform cloud observed in the Hulun Buir region in the northeast of China on 3 August 2023.**

455

To determine the AgI particle concentration, firstly we need to choose a threshold of Ze induced by cloud seeding, i.e., the seeding-induced Ze should be higher than this threshold thus the composite Ze (natural and seeding) can exceed the natural variability after seeding. Figure 10 shows the contoured frequency-by-altitude diagram (CFAD) of observed Ze, the solid red line indicates the average Ze plus one standard deviation, and the dashed line indicates the mean plus 2 standard deviations.

460 These two thresholds are probably still not large enough if the seeding is performed in regions with relatively high natural Ze (e.g., at 16:40), but may be fine if the seeding is conducted at 16:10 when the natural Ze is low. To better detect the seeding signature, a larger value of Ze is necessary, such as the mean plus 3 standard deviations (dotted line), which well exceeds the maximum natural Ze.

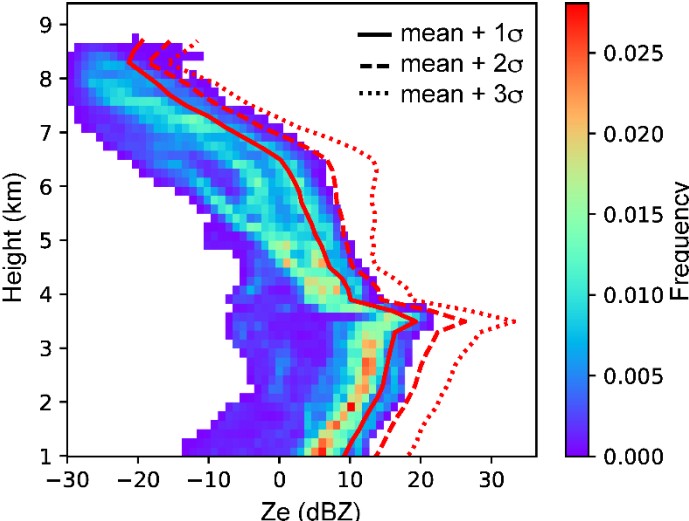

465 **Figure 10. CFAD of Ze observed by the Ka-band radar for the period shown in Fig. 9b. The solid, dashed, and dotted lines indicate the mean Ze plus 1, 2 and 3 standard deviations, respectively.**

## 4.2 AgI concentration needed to detect the seeding signature

Figure 11 shows the AgI particle concentration that is needed at different heights (-22 °C - -7 °C) to detect the Ze attributed to seeding above the freezing level (4.2 km), assuming there is sufficient supercooled liquid water to support the ice growth. We test three different thresholds for seeding-induced Ze (5.9 dBZ, 11.2 dBZ, and 16.7 dBZ). These values correspond to a composite Ze (mean natural Ze and seeding Ze) of 8 dBZ, 12 dBZ, and 17 dBZ above the freezing level (see the three red lines in Fig. 10). It is within the expectation that more AgI is needed to obtain a larger Ze, but the vertical variations of the three profiles of AgI particle concentration are very similar (Fig. 11). The required AgI concentration is the lowest at about 7.3 km (-15 °C), where 20 cm$^{-3}$ AgI is sufficient for the cloud seeding to induce a composite Ze exceeding 17 dBZ. Therefore, seeding at this level provides the highest probability of detecting the seeding signature. However, airborne seeding at -15 °C is not always possible because of flight limitations (e.g., airframe icing). Seeding at colder or warmer temperatures may also exceed the Ze threshold if the AgI particle concentration is high enough, however, since a normal AgI concentration in cloud seeding operation has a magnitude of 10-1000 cm$^{-3}$, it is unlikely that seeding below 6.5 km will yield a detectable seeding signal near the freezing level in this case.

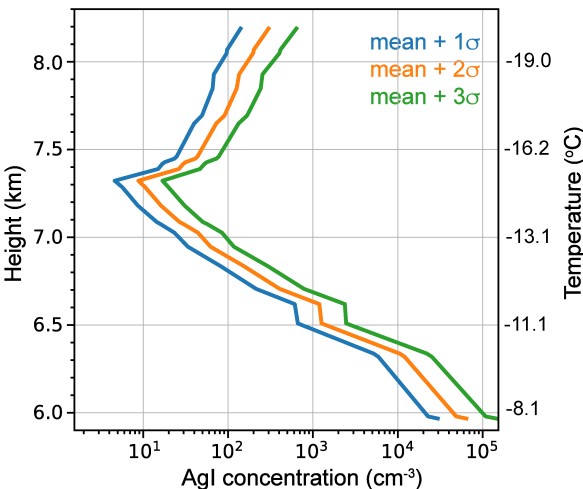

**Figure 11. The profiles of AgI concentration needed to detect the seeding signature at the freezing level for different Ze anomaly thresholds.**

In precipitating clouds, the presence of natural ice would diminish the detectability of the seeding signature because the natural and seeded ice crystals would compete for the supercooled liquid water. We do not consider the interaction between the seeded and natural ice in the 1D model because we assume there is sufficient supercooled liquid water for their growth, and aggregation between natural and seeded ice is expected to further enhance the ice crystal size, which is favorable for seeding signature detection. Therefore, we are able to apply the parametrization to a precipitating cloud, at least, it provides a threshold

of AgI particle concentration that is needed to detect unambiguous seeding signatures. However, the assumption of sufficient liquid water is not always valid, especially when the natural ice concentration is high. Figure 12 shows the liquid water path (LWP) between the seeding level and the target level (4.2 km), needed to support the growth of ice for different seeding temperatures. Seeding at -15 °C consumes the least liquid water, which is less than 30 g m$^{-2}$. The LWP in this cloud observed by a microwave radiometer was mostly larger than 300 g m$^{-2}$, however, both rain water and cloud water contribute to the LWP,

we do not have direct measurements of LWC in clouds, so it is not known whether there is sufficient water for the ice growth if seeding at a temperature of -21 °C – -11 °C. Seeding at temperatures warmer than -10 °C requires much more liquid water due to the high AgI particle concentration, therefore, the size of ice crystals and Ze may be even smaller as suggested by the model. Previous observational studies suggest the LWP in shallow mixed-phase stratiform clouds is often lower than 100 g m$^{-2}$ (Zhang et al., 2014). Such a low LWP prevents the growth of ice crystals if the ice concentration is high. In addition, if there

is no continuous supply of liquid water (e.g., in orographic updrafts), seeding can cause complete cloud glaciation in the seeded region (Dong et al. 2021).

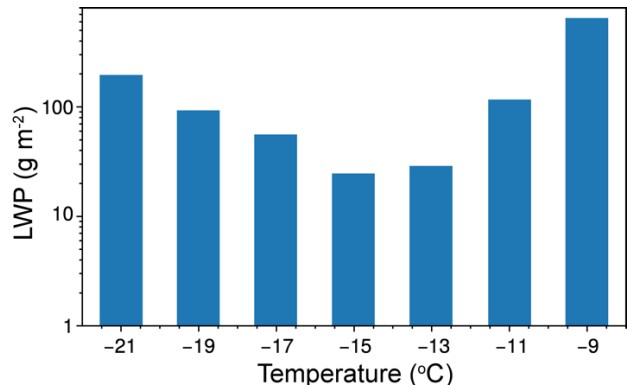

**Figure 12. The LWP needed to support the growth of ice for different seeding temperatures between the seeding level and the target level.**

**5 Discussion and Conclusions**

In this study, the IWC and Ze induced by glaciogenic seeding using different AgI particle concentrations under various cloud conditions are investigated using a 1D ice growth model coupled with an AgI nucleation parameterization. In addition, an algorithm is developed to estimate the AgI particle concentration that is needed to detect the signature of glaciogenic cloud seeding. This algorithm, which is a parameterization that links the AgI particle concentration to the radar Ze induced by seeding,

is developed based on multiple numerical experiments using the 1D model. The main conclusions are as follows:

(1) The 1D model captures characteristics of the ice growth habit compared to laboratory experiments, and the modelled IWC and Ze are consistent with the 3D LES model. However, the evaluation was conducted using only two case studies, further model validation is needed in the future, especially using field measurements.


(2) Since ice crystals have the largest growth rate at -15 °C, the IWC and Ze induced by cloud seeding at -15 °C is larger than seeding at colder temperatures. Therefore, a radar seeding signature is most likely to be detected at a temperature of about -15 °C. This finding is consistent with the fact that the seeding temperature was about -15 °C or slightly warmer in most documented unambiguous seeding signature cases.


(3) A higher AgI particle concentration leads to higher Ze assuming all other parameters are equal. Both the 1D and 3D models suggest that for an AgI particle concentration of 35 cm$^{-3}$, the Ze induced by cloud seeding is 10-20 dBZ, and for an AgI particle concentration of 350 cm$^{-3}$, the Ze is 20-30 dBZ, as long as sufficient liquid water is available.

(4) Using the 1D model, 2500 numerical experiments were conducted for various cloud conditions and AgI particle concentrations. Statistical analysis based on the model data indicates AgI particle concentration and temperature are the major factors controlling the IWC and Ze at the target level. The depth from the seeding level also influences the IWC and Ze as a deeper cloud provides a longer path for ice growth.

(5) Based on the data from the 2500 numerical experiments, a parameterization is developed using polynomial regression to estimate the minimum AgI particle concentration that is needed to detect the signature of glaciogenic cloud seeding, assuming there is sufficient liquid water for ice growth. The seeding temperature, pressure, Ze, and cloud depth from the seeding level are used as the inputs to train the parameterization, and the AgI particle concentration is the output.

(6) Application of this parameterization to a real case with natural precipitation suggests seeding at about -15 °C requires the least AgI to obtain a seeding signature exceeding the natural variability. Seeding at slightly colder or warmer temperatures also may produce a detectable signature, but requires more AgI and supercooled liquid water. Seeding impact at temperatures warmer than -10 °C is unlikely to be detected in this case, because it requires an extremely high concentration of AgI and an LWP exceeding 600 g m$^{-2}$. However, for nonprecipitating clouds, seeding at temperatures warmer than -10
°C may produce detectable seeding signature.

This study has several limitations:
- The results shown here only apply to mixed-phase stratiform clouds with relatively simple dynamics, because we assumed a relatively weak turbulence in the model. In clouds with stronger turbulence or convections, the ice growth trajectories

545        are more complicated, thus, a 3D model has to be used to investigate the seeding-induced signature under various ambient conditions (Xue et al., 2022; Hua et al., 2024).

-      AgI released pyrotechnically from point sources on the ground or in the air (burn-in-place or ejectable flares, rockets …) vary vastly in concentration. In fact, AgI particle concentrations in 3D models typically are depicted on a log scale rather than a linear scale (e.g., Xue et al. 2013b; 2022). Therefore, seeding operators have little control over AgI concentrations.

But as AgI disperses from a point source in boundary layer or cloud turbulence, our study shows that it may cross a "sweet spot" where, under given cloud conditions, the seeding impact is optimally detectable.

-      The parameterization developed in this study applies to airborne cloud seeding. For ground-based seeding, which is often used for orographic clouds, the AgI particles are mixed into clouds mainly through boundary layer turbulence. Boundary layer convection and hydraulic jump in the lee may enhance the vertical dispersion of AgI particles (Jing et al., 2016).

Therefore, to investigate the ice nucleation and growth, it is important to resolve the vertical dispersion of AgI particles (Xue et al., 2013b), which is not considered in the 1D model presented in this study. High-resolution LES model is the better choice to model the impact of ground-based cloud seeding (Xue et al., 2013b; Chu et al., 2014).

-      In our study, it is expected that cloud seeding contributes the majority of ice mass in the seeding plume. The best radar seeding signatures come from clouds with no or very weak natural precipitation (Friedrich et al., 2020). In reality, natural

snowfall may contribute significantly. It is difficult to quantitatively separate the precipitation attributed to cloud seeding from natural precipitation. Nevertheless, unambiguous seeding signatures in precipitating clouds can still be useful in studying the chain of physics of cloud seeding.

-      The 1D model does not thermodynamically constrain the LWC available for glaciogenic seeding. Under limited LWC, a larger AgI concentration may reduce the radar reflectivity enhancement, because more, but smaller ice crystals will form, compared lower AgI concentration. Such constraint is captured in a 3D model.


In short, the results shown in this study deepen our understanding of the relationships between AgI particle concentration and Ze under different cloud conditions. The parameterization has limitations, but it can be useful in seeding operations to provide a quick estimation of how much AgI particle concentration is needed to obtain an unambiguous seeding signature.


**Data availability**

The WRF model is available on https://www2.mmm.ucar.edu/wrf/users/download/get_source.html (NCAR MMM, 2023). The sounding data, radar data and ice growth model are available on https://doi.org/10.5281/zenodo.12793527 (Yang, 2024).

## Author contributions

JY, JL, and MC conducted the numerical simulations. JY, JL and XJ evaluated and analyzed the model results. JY and XJ prepared the paper. HH provided the data of radar measurements. YY, BG, ZW, YL and BC provided inputs on the method and analysis. All the authors provided significant feedback on the paper.

## Competing interests

The contact author has declared that none of the authors has any competing interests.

## 580 Acknowledgements

This work was supported by the National K&D Program of China (2023YFC3007600), the National Natural Science Foundation of China (42475201, 42230604), and the CMA Key Innovation Team Support Project (CMA2022ZD10). We acknowledge the High Performance Computing Center of Nanjing University of Information Science & Technology for their support of this work. We appreciate the editor and reviewers for their insightful comments and suggestions.

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
