# Peer review of "Estimating the concentration of silver iodide needed to detect unambiguous signatures of glaciogenic cloud seeding"

_EGUsphere, 2024_

## Referee Comment (RC2)

**Title:** Estimating the concentration of silver iodide needed to detect unambiguous signatures of glaciogenic cloud seeding

**Manuscript Number:** egusphere-2024-2301

**Recommendation:** Accept with major revisions

Yang et al. (2024) examines the optimal AgI concentrations for cloud seeding to produce detectable radar signatures in mixed phase stratiform clouds. Using a 1D ice growth model with an AgI nucleation parameterization, the authors conducted 1000 simulations to explore how factors like temperature, pressure, and seeding height influence the reflectivity response/detectability. Findings indicate that seeding is most effective around -15°C, which requires the least AgI for detectable signatures, while seeding at colder or warmer temperatures demands higher concentrations of supercooled water for the same response. Although the 1D model captures key ice growth characteristics, it simplifies cloud dynamics and assumes uniform conditions, limiting applicability. Comparisons with 3D LES simulations highlight these limitations near the seeding level. The parameterization offers a tool for estimating poretnail airborne AgI seeding contribution but is less applicable to ground based seeding which is more dependent on different dispersion mechanisms. Although I think the manuscript is novel in showing optimal conditions for the detectability of seeding given varying seeding material amounts and is well written with few grammatical errors, I do have several suggestions that would help improve the quality of the manuscript.

**General Comments:**

The paper needs to clearly specify the vertical temperature profile used throughout the study and how this profile relates to the assumed background liquid water content (LWC). The background LWC in the analysis is estimated by assuming an adiabatic cloud, but the details of this assumption are not fully explained. I suggest including a figure showing the temperature profile and corresponding LWC for all 1D simulations earlier in the study to help the reader understand the background conditions and the baseline environment in which the seeding effects are modeled.

Clouds in real-world conditions rarely exhibit perfectly adiabatic liquid water content profiles. It is important for the authors to clearly state that the assumption of an adiabatic profile represents a theoretical upper limit of liquid water content available for seeding. If clouds contain less supercooled liquid water, it is likely that more AgI would be required to achieve comparable seeding effects. To address this, the paper would benefit from a brief sensitivity analysis that explores how variations in background SLW content impact the amount of AgI needed to produce similar reflectivity enhancements. Such an analysis would provide a more realistic comparison and better represent the variability found in natural cloud conditions.

The study assumes that all clouds examined were initially composed entirely of supercooled liquid water with no background ice crystals. This assumption significantly impacts the interpretation of the seeding effects, as introducing ice crystals in an ice-free environment is likely to produce substantial changes in reflectivity and ice water content. In natural mixed-phase clouds, where ice crystals already exist, the reflectivity changes resulting from seeding may be much more subtle

due to the higher baseline of ice content. Therefore, the large reflectivity increases reported in this study could be partially attributed to the idealized initial conditions rather than the inherent efficacy of seeding under more typical, mixed-phase cloud conditions. This raises questions about the generalizability of the findings to real-world scenarios, where the presence of natural ice would likely diminish the detectability of the seeding signature. The authors need to explicitly state this assumption especially in the abstract and conclusions and add a statement at the end on its implications for the interpretation of their results. This is especially true for Section four, where the study attempts to translate the impact of adding additional AgI in a completely supercooled cloud to scenarios with background natural IWC and likely (but unknown) variable supercooled liquid water conditions. The authors need to explicitly state their justification at the beginning of the section (there is some justification at the end) on why they can make this translation to clouds with natural background ice populations.

The paper's conclusions heavily rely on a single comparison with 3D LES data, which raises concerns about the robustness and generalizability of the findings in Sec. 3.2. The 1D simulations appear to involve seeding in a deeper orographic cloud (up to 6 km) over several kilometers, whereas the LES comparison is conducted on a much shallower stratiform cloud, only about 600 m thick, at a single temperature. This discrepancy in cloud depth and structure between the 1D and 3D simulations could significantly influence the modeled seeding effects, particularly in terms of ice growth dynamics, particle fall speeds, and reflectivity changes. Additionally, the thermodynamic environments likely differ between the 1D model runs and the LES simulation, including variations in temperature, moisture profiles, and turbulence intensity, which are critical factors influencing ice nucleation and growth. Without a detailed comparison of these environmental differences, the validity of using the LES as a representative or benchmark for the 1D results is questionable. The paper would benefit from a more thorough examination of how these thermodynamic discrepancies might affect the outcomes to ensure the conclusions are not overly reliant on a single, potentially unrepresentative LES case.

The study attributes reflectivity enhancement primarily to particle growth, but it does not adequately disentangle the contributions from nucleation or increased particle growth versus dispersion effects. A deeper analysis into how much of the observed reflectivity increase due to ice crystal growth, changes in particle size distribution, or enhanced dispersion of particles within the 1D model would provide a clearer understanding of the seeding impacts. This would be nice to show for a single simulation run. Such analysis would clarify the main drivers of reflectivity enhancement and guide more targeted seeding strategies.

**Specific Comments:**

Line 126: Was scavenging of AgI by nucleation accounted for by the 1D model? That one was of the parameterizations in Xue et al. 2013a.

Line 126: This would be an appropriate place to discuss seeding strategy, which was limited in the manuscript. Did seeding occur at the specified temperature height? How long did seeding occur? Was it for the entire period of the 1D simulation?

Line 135: Suggest rewording the second half of the sentence: "Radiative cooling at cloud top reduces stability resulting in weak vertical motions at cloud top that may also enhance supersaturation."

Line 146: Should this be temperature and supersaturation dependent?

Line 182: Suggest rewording to: terminal fall velocity.

Sec. 2.2: What was the vertical temperature profile used in this analysis? Was it consistent for all 1000 simulation runs?

Sec 2.2: A statement needs to be added to this section that the 1D model does not incorporate aggregation or secondary ice production mechanisms, such as rime-splintering or Hallett-Mossop processes, which can significantly alter ice particle size distributions and radar reflectivity. By not including these processes, the model may underestimate the reflectivity in conditions where aggregation is a dominant growth mechanism, or in clouds with abundant SLW or high ice crystal concentrations as a result of SIP.

A statement also needs to be added clarifying that nucleated particles that form at a given level are either columnar or plate-like depending on background temperature and remain that way as they descend to the surface.

Sec. 2.4: A statement needs to be added to this section discussing the uncertainties when making radar reflectivity calculations using the assumption of Rayleigh scattering, which is valid for larger particles and longer wavelengths but may not hold at shorter wavelengths (e.g., W-band radar) where reflectivity becomes more sensitive to particle concentration rather than size. This assumption could limit the parameterization's applicability across different radar systems used in cloud seeding operations. A similar statement can be found in the conclusions.

Fig. 2: Why are there mean layers where particles consistently see decreases in IWC and $Z_e$, if particles are growing with depth? How random is the turbulent kinematic diffusion between the different runs? Are their changes in LWC as a result of variations in the temperature profile under the assumption that the cloud is adiabatic? A figure showing initial conditions would be useful to add at the beginning of the analysis.

Fig 2: A panel showing total number concentration for each distribution would be helpful to the reader to determine how many ice crystals are contributing to $IWC$ and $Z_e$ at a given height.

Fig. 6: Note in the caption that you are only comparing the -15°C simulation with these LES results in the figure caption.

Figure 7: I found Fig. 7 difficult to interpret because there are four parameters. Could it be broken into a three panel plots showing temperature vs AgI, temperature vs height, and then AgI vs height with reflectivity shaded as a function of reflectivity enhancement. That may make it easier to interpret.

Line 272: Suggest removing the sentence starting with "Long-term remote sensing measurements". It seems out of place and doesn't add anything to the discussion without showing additional data.

Line 433: This assumption needs to be clearly stated at the beginning of this section.

---

## Author Comment (AC1)

The reviewer's comments are in black, and responses are in blue.

The authors present a study that develops a 1D model aimed at predicting radar reflectivity changes resulting from glaciogenic cloud seeding using silver iodide (AgI). The primary goal of the study is to determine the AgI concentrations required to generate an unambiguous seeding signature, allowing for the detection of the effects of cloud seeding on radar. To achieve this, the authors conducted 1000 simulations under varying AgI concentrations and temperature conditions. The findings indicate that the lowest AgI concentration required to produce a detectable seeding signal occurs at -15°C, while at temperatures above -11°C, a signal is only observed in precipitating clouds when both high AgI concentrations and supercooled liquid water are present. The study's results were evaluated using 3D model simulations and observational data, providing insights into the behavior of seeded clouds.

The manuscript is generally well-written and falls within the scope of the journal. The model presented is a valuable contribution and has the potential to aid in the planning of seeding missions. The parameterizations and methods are, for the most part, well described. However, several areas require clarification, particularly concerning the choice of parameterizations and some missing information crucial for fully understanding the model's behavior. While the structure of the paper is solid, a more in-depth discussion of specific modeling assumptions and a stronger comparison with existing literature would enhance the clarity and scientific impact of the work.

Jan Henneberger

Reply: Dear Dr. Henneberger, we appreciate your insightful comments. The paper has been revised accordingly and has been improved a lot. All the figures have been updated because we now use the ice crystal growth ratios parametrized by Harrington et al. (2019) according to your suggestion. Please see our point-by-point response below.

Major comments:
A critical issue that needs addressing is the distance to the seeding when discussing AgI particle concentrations. Given that a single seeding flare can release approximately 1020 AgI particles, one would expect much higher concentrations near the seeding

source. It would be helpful to clarify at what distance the concentrations mention in the manuscript are assumed.

Reply: We appreciate your comment. In our 1D and 3D models, the concentration of AgI particles mentioned in the text is exactly at the seeding level. This is now clarified in the paper. We agree that AgI particle concentration could be higher near the source than its surrounding areas. The parametrization given in this paper provides a "threshold" of AgI particle concentration to detect unambiguous seeding signals. The required AgI particle concentration could either be at the seeding level or at some distance below the level. If one wants to estimate the AgI particle concentration at some distance below the seeding level, a vertical dispersion model is needed, and we also need to know how the particles are released (e.g., rockets, aircraft, etc..), and how many seeding flares are used at the same time.

The manuscript does not provide sufficient detail on the assumed temperature profile used in the 1D model. The authors should clarify whether the results are sensitive to the temperature profile and how different profiles might impact the outcome of the simulations.

Reply: We appreciate your comment. In the model, the temperature profile is determined by the cloud top temperature and lapse rate. Once the cloud top temperature is given, we can calculate the temperature at different levels. Typically, the lapse rate in a stratiform cloud is 5-6 K/km, we made sensitivity tests using different lapse rates, and an example is shown in Fig. R1. It is seen that the results are quite similar using a lapse rate of 5 and 6 K/km. In this study, we use 5.5 K/km, and a random temperature perturbation varies between -0.1 K and 0.1 K is applied to each level. Sorry for missing the information about temperature, this is added to the revised paper.

[Figure]

Figure R1. Vertical profiles of Ze from the simulations with a temperature lapse rate of (a and b) 5 K/km, and (c and d) 6 K/km, and an AgI particle concentration of (a and c) 35 cm$^{-3}$, and (b and d) 350 cm$^{-3}$.

The current comparison of the 1D model with the 3D models and with observational data is limited. A single comparison is insufficient to draw robust conclusions, and further comparisons would strengthen the validation of the 1D model.

Reply: We appreciate your comment. In the revised paper, we compare the 1D model with the 3D model using 2 cases, a shallow cloud (same as the case in the original manuscript), and a deeper one. The sounding data is originally based on the shallow case (solid line in Fig. R2). To model the deeper case, we modify the temperature and vapor mixing ratio data (dashed line in Fig. R2). The deeper case has a cloud depth of 2 km, seeding is performed at -21 °C. In the shallow case, seeding is performed at -15 °C. The other model configurations are the same as the original manuscript. We acknowledge that this model and the parameterizations need more validation in the future, especially using observational datasets such as Cloudlab.

[Figure]

Figure R2. The initial profiles of (a) temperature and potential temperature, (b) vapor mixing ratio, and (c) U and V components of the wind field. The solid lines indicate the original data used for the shallow cloud, and the dashed lines indicate the modified data used for the deep case. The dark-shaded area (1.3km – 1.9 km) indicates the shallow cloud layer and the light-shaded area (1.3 km - 2.9 km) indicates the deep cloud layer.

Figure R3 shows the modelled Ze, and IWC. It is seen that the Ze and IWC are much greater for the shallow case (Fig. R3a-d) than the deeper one (Fig. R3e-h), because seeding is conducted at -15 °C in the shallow case, while at -21 °C at the deeper case.

[Figure]

Figure R3. Cross sections of Ze (rainbow shading), liquid water mixing ratio (blue shading), and ice mixing ratio (contoured) were obtained from 3D LES simulations for (a-d) the shallow cloud, and (e-h) the deeper case.

Statistically, the 1D model is consistent with the 3D model. Although there are inevitable differences (a few dBZ) between the 1D and 3D model. Here, we would like to point out there was a mistake (incorrect height data) when plotting the 3D model results in Fig. 6. In the new version, we found the 1D model and 3D model shows similar vertical variation (Fig. R4).

[Figure]

Figure R4. The vertical profiles of Ze simulated using the 1D and 3D models with an AgI particle concentration of (a, c) 35 cm$^{-3}$, and (b, d) 350 cm$^{-3}$ for the (a, b) shallow and (c, d) deep cases, respectively. The left and right boundaries of the blue shaded area indicate the 95th percentile and the maximum Ze from the 3D model, respectively.

One notable omission is the lack of citation to recent relevant studies, including our own work (Henneberger and Ramelli et al., 2023). I find myself in the somewhat awkward position of promoting my own research, which I assume was not cited because it was published only recently. However, I believe it is highly relevant to the current manuscript. In our study, we observed a clear seeding signal in over 50 missions, with ice crystal concentrations reaching up to 1000 L$^{-1}$. This demonstrated the potential to detect unambiguous seeding signatures even at temperatures as high as -5°C, given favorable background conditions and sensitive instrumentation. Additionally, our Cloudlab dataset would provide a strong test case for the model presented here, though this might be outside the immediate scope of the current paper.

Reply: We appreciate your comment, and are sorry for missing such an important reference. In addition, we notice a new article by Ramelli et al. (2024, PNAS NEXUS), which is also quite relevant to this study. The following statements are added to the manuscript:

"Recently, by seeding supercooled stratus cloud with an uncrewed aerial vehicle, Henneberger et al. (2023) provide new observational evidence of precipitation enhancement at temperatures as high as -5°C. Unambiguous seeding signature was detected using in-situ and ground-based remote sensing instruments when the background noise is low".

"It should be noted that uncertainties are inevitably present in modelling ice growth in both the 1D model and the 3D models used in this paper. Improving the ice growth parameterizations in model is vital for the purpose of this study. Recently, using measurements in seeded clouds, Ramelli et al. (2024) showed the ice growth rates have large variabilities, indicating that in real clouds the dynamics and microphysics are complicated."

The Cloudlab dataset is unique to evaluate the microphysics schemes, and can be used to investigate the ice growth and nucleation in real clouds. It would be very interesting to use this dataset to test models, not only ours, but also the various 2D and 3D models.

Have you considered including the linear depolarization ratio to detect the seeding signal? It may offer higher sensitivity, especially for ice crystals with large aspect ratios, and with polarization radars becoming more common, this could be a valuable addition.

Reply: We appreciate your comment. We agree, it would be worth of investigating the signature of the linear depolarization ratio induced by cloud seeding. However, in our opinion, this work needs much more effort and is more appropriate for a separate paper. Firstly, we need measurements of linear depolarization ratio in natural and seeded clouds, either shown in previous studies or by ourselves, to test the 1D model, however, to our knowledge, such information is quite limited (Henneberger et al., 2023; Jing et al. 2015). Secondly, to test the performance of the 1D model, we compare it with WRF 3D model, while the linear depolarization ratio cannot be modelled using WRF, extensive efforts need to be made to modify the WRF model code to get the linear depolarization ratio, including the representation of ice crystal shapes, which is out of the scope of this paper. Thirdly, the primary purpose of cloud seeding is to enhance precipitation, which is more related to ice concentration and size. One may imagine that

if natural ice and seeded ice crystals form at the same level, they may have similar linear depolarization ratio, but the concentration of seeded ice is higher. Linear depolarization ratio has the potential to better identify seeding signatures. For example, if we can measure the background linear depolarization ratio, we can decide at which level to seed, thus ice shapes induced by seeding are different from the natural ice (i.e., different linear depolarization ratios). This discussion is added to the revised manuscript.

Minor comments:

Line 104: Have you thought about using the Marcolli et al., 2016 parameterization for ice nucleation after Omanovic et al., 2024 showed that the DeMott (1995) parameterization was not active enough at warm temperatures.

Reply: We appreciate your comment. Recently, we evaluated three seeding parameterizations in two different microphysics schemes (Hua et al., 2024). The results suggest the differences in the seeding effect induced by the three seeding parametrizations are smaller than those by different microphysics schemes. However, Marcolli et al., 2016 parameterization is not tested in Hua et al., (2024), it would be interesting to test this parameterization in the future, but this needs to be tested in both the 1D and 3D models. The accuracy of ice nucleation is definitely a source of uncertainty in the model, we have added this discussion in the revised paper.

Reference:

Hua, S., B. Chen, H. He, Y. Chen, X. Liu, and J. Yang, Numerical simulation of the cloud seeding operation of a convective rainfall event occurred in Beijing. Atmos. Res., 2024, 304, 107386, https://doi.org/10.1016/j.atmosres.2024.107386.

Line 133: The WBF process is also not active in stronger downdrafts, as both ice crystals and droplets shrink. I would rather argue that weak turbulence cancels out the effects of updrafts and downdrafts.

Reply: We agree. Both ice crystals and droplets shrink if the downdraft is strong. Previous studies show that turbulence do have an impact on snow growth (e.g., Chu et al., 2018). Therefore, stating that weak turbulence cancels out the effects of updrafts and downdrafts maybe inappropriate. In the revised paper, we simply add "weak" before "downdraft" to emphasize the model applies to weak turbulence (i.e., weak updraft and weak downdraft).

Reference:

Chu, X., Xue, L., Geerts, B., and Kosovic, B., The impact of boundary layer turbulence on snow growth and precipitation: Idealized Large Eddy Simulations, Atmospheric Research, 2018, 204, 54-66, https://doi.org/10.1016/j.atmosres.2018.01.015.

Line 139: Clarify how you calculate ice supersaturation ($S_i$) during the simulation. Does $S_i$ depend on the liquid phase or turbulence?

Reply: We assume the WBF process is always taking place during the simulation, so $S_i$ depends on the liquid phase and temperature.

Line 143: A X is missing in equation 6.

Reply: Thank you for the comment. X is added to equation 6.

Line 165: Ice crystal growth ratios are currently fit to one laboratory study. Consider using Harrington et al. (2019), which take multiple laboratory measurements into account.

Reply: We appreciate your comment. We now use the ice crystal growth ratios parametrized by Harrington et al. (2019), it seems that the ice mass modelled using this parametrization is slightly better than the original one (Fig. R5). All the figures have been updated.

[Figure]

Figure R5. (a) Growth of the mass of a single ice crystal as a function of time at different temperatures under a standard atmospheric pressure. The dots are from laboratory experiments conducted by Takahashi et al. (1991) and the curves are from model simulations. (b) and (c) are similar to (a) but for the a-axis and c-axis, respectively.

Line 226: How does dispersion in the 1D model compare to the 3D model? I would expect greater dispersion in the 3D model due to the extra dimensions.

Reply: We appreciate your comment. Yes, the dispersion in the 3D model is slightly greater than in the 1D model, especially in the vertical direction. In the horizontal direction, we tested the sensitivity of Ze to the dispersion coefficient. As seen in Fig. R6, the dispersion would cause an uncertainty of a few dBZ. This figure, as well as some other sensitivity tests, are added to the revised paper.

[Figure]

Figure R6. Vertical profiles of Ze simulation using different turbulence dispersion coefficients.

Line 257: What altitude was seeding performed on, and how was the vertical temperature profile set? What is the cause of the variation in IWC with altitude? I would have expected a linear increase as altitude decreases.

Reply: In this study, the seeding level is not constant. For example, in Fig. 2 in the paper, seeding is performed at different heights. The vertical variation of IWC is because the ice concentration varies with height. However, we made a mistake in generating the random numbers of turbulence. This is corrected in the code and the figure is updated (Fig. R7). In the updated figure, we still see a slight fluctuate variation of IWC, because the ice concentration is determined by the ice nucleation rate parameterized by Xue et al. (2013), and it is related to the turbulent vertical velocity, random numbers in the turbulent vertical velocity cause the variations of ice concentration.

[Figure]

Figure R7. Vertical profiles of (a and d) ice concentration, (b and e) IWC, and (c and f) Ze from the simulations with different seeding temperatures and an AgI particle concentration of (a-c) 35 cm$^{-3}$, and (d-f) 350 cm$^{-3}$. The results are obtained based on 10 numerical experiments for each seeding temperature. The shaded area captures the 20th-80th percentile range, and the solid lines are the mean profiles.

Line 261: Ice crystal concentrations seem to depend only on temperature and seeding concentration and are calculated at the start of the simulation. Please clarify if these values change over time.

Reply: Ice crystal concertation is also determined by ice nucleation rate, so it is related to turbulence and changes with time.

Line 287: What was the seeding height?

Reply: The seeding is performed at 1.8 km (about -15 °C).

Line 296: Provide details on the spatial dimension and temporal duration of the seeding?

Reply: Seeding is simply conducted at a single time (30 min), along a straight line across the domain in the south-north direction, which has a length of 10 km.

Figures 4, 5, 6: Indicate the seeding height and clarify whether heights are measured above ground or sea level. In Figure 4, explain why the seeding signal is closer after 30 minutes than after 20 minutes.

Reply: The seeding height is at 1.8 km above the sea level. This information is added to the paper. Thank you for pointing out the difference between 20 min and 30 min results. This is probably because, in a short time, only some of the snow crystals can fall out of the cloud. We made a sensitivity test using the 1D model. As seen in Fig. R8, most of the ice is above 1.3 km after seeding 15 min, only a small fraction of snow crystals can reach a lower height due to their larger terminal velocities. After 20 min, the seeding signal can extend to lower levels. After 25 min, the Ze profile becomes stable, i.e., Ze no longer changes with seeding time in the cloud layer (assuming there is a continuous liquid water supply). However, the dynamics are simplified in the 1D model, and we did not consider the vertical mixing of ice crystals, it is expected that the Ze within 20 min would be a few dBZ smaller above the cloud base if more realistic turbulence and vertical mixing is considered (like 3D model).

[Figure]

Figure R8. Vertical profiles of Ze from the simulations with different seeding time durations.

Line: 355: Specify the temperature profile used. Why are T_seed and P_seed needed at independent input?

Reply: The temperature profile is determined by the cloud top temperature and lapse rate. Please see our response to your Comment 1. T_seed and P_seed are in dependent because low temperature can be observed either at low altitude (high pressure) or high altitude, this may be caused by seasonal variation or latitudinal variation. For example, a cloud with a top temperature of -15 °C in winter is observed at a higher pressure than in summer.

Line 358: At what distance from the seeding source are AgI concentrations assumed? Higher concentrations should be expected near the flare.

Reply: The AgI concentration is at the seeding level.

Line 366: Can radar reflectivity (Ze) decrease with increasing depth if ice crystals cannot shrink in the 1D model?

Reply: In our 1D model, Ze generally increases with increasing depth. But Ze not only depends on ice crystal size, it also depends on ice concentration. The fluctuation of Ze is controlled by the fluctuation of ice concentration (e.g., Fig. R7), which is related to the ice nucleation rate.

Line 385: Data must be shown if it is discussed. Also, varying the number of experiments (e.g., 500 and 2000) would provide more informative results.

Reply: We appreciate your comment. In the revised paper, we ran 2500 experiments, 2000 of them were used to train the polynomial regression, and the remaining 500 were used for testing. The results are shown in Fig. R9, and the conclusions remain similar.

[Figure]

Figure R9. (a) Scatter plot of the parameterized and modelled AgI particle concentration. The grey dots are the results from the 1000 numerical experiments for training, and the dark blue dots are binned averages. The parameterization is developed using the polynomial regression with a degree of 6. (b) Similar to (a) but for the 500 experiments used for the test. (c) RMSE and correlation coefficient between the parameterized and modelled AgI concentration for different polynomial degrees used in the regression of training data.

Line 437: Change to "was mostly larger"

Reply: "is mostly larger" is changed to "was mostly larger" in the revised paper.

Line 438: Are you certain there is sufficient water available for ice growth below -13°C? Based on the radar reflectivity, most of the liquid water content (LWC) appears to be concentrated below the melting layer.

Reply: We agree, we do not have direct evidence to show the LWC is sufficient. This sentence is removed from the manuscript.

Line 456: State that the evaluation was conducted using only one case study.

Reply: We now evaluate the model using two cases and this statement is added in Section 5.

Line 465: Replace "ceteris paribus" with "all other parameters being equal" for better readability.

Reply: "ceteris paribus" is replaced with "all other parameters being equal".

References:

Harrington, J. Y., A. Moyle, L. E. Hanson, and H. Morrison, 2019: On Calculating Deposition Coefficients and Aspect-Ratio Evolution in Approximate Models of Ice Crystal Vapor Growth. J. Atmos. Sci., 76, 1609–1625, https://doi.org/10.1175/JAS-D-18-0319.1.

Henneberger, J., Ramelli, F., and Coauthors, 2023: Seeding of Supercooled Low Stratus Clouds with a UAV to Study Microphysical Ice Processes: An Introduction to the CLOUDLAB Project. Bull. Amer. Meteor. Soc., 104, E1962–E1979, https://doi.org/10.1175/BAMS-D-22-0178.1.

Marcolli, C., Nagare, B., Welti, A., and Lohmann, U.: Ice nucleation efficiency of AgI: review and new insights, Atmos. Chem. Phys., 16, 8915–8937, https://doi.org/10.5194/acp-16-8915-2016, 2016

Omanovic, N., Ferrachat, S., Fuchs, C., Henneberger, J., Miller, A. J., Ohneiser, K., Ramelli, F., Seifert, P., Spirig, R., Zhang, H., and Lohmann, U.: Evaluating the Wegener–Bergeron–Findeisen process in ICON in large-eddy mode with in situ observations from the CLOUDLAB project, Atmos. Chem. Phys., 24, 6825–6844, https://doi.org/10.5194/acp-24-6825-2024, 2024

---

## Author Comment (AC2)

The reviewer's comments are in black, and responses are in blue.

Yang et al. (2024) examines the optimal AgI concentrations for cloud seeding to produce detectable radar signatures in mixed phase stratiform clouds. Using a 1D ice growth model with an AgI nucleation parameterization, the authors conducted 1000 simulations to explore how factors like temperature, pressure, and seeding height influence the reflectivity response/detectability. Findings indicate that seeding is most effective around -15°C, which requires the least AgI for detectable signatures, while seeding at colder or warmer temperatures demands higher concentrations of supercooled water for the same response. Although the 1D model captures key ice growth characteristics, it simplifies cloud dynamics and assumes uniform conditions, limiting applicability. Comparisons with 3D LES simulations highlight these limitations near the seeding level. The parameterization offers a tool for estimating potential airborne AgI seeding contribution but is less applicable to ground based seeding which is more dependent on different dispersion mechanisms. Although I think the manuscript is novel in showing optimal conditions for the detectability of seeding given varying seeding material amounts and is well written with few grammatical errors, I do have several suggestions that would help improve the quality of the manuscript.

Reply: We appreciate your insightful comments. The paper has been revised accordingly, and has been improved a lot. In addition, according to reviewer 1's comment, we updated the ice growth model, and all the figures have been updated. Please see our point-by-point response below.

General Comments:

The paper needs to clearly specify the vertical temperature profile used throughout the study and how this profile relates to the assumed background liquid water content (LWC). The background LWC in the analysis is estimated by assuming an adiabatic cloud, but the details of this assumption are not fully explained. I suggest including a figure showing the temperature profile and corresponding LWC for all 1D simulations earlier in the study to help the reader understand the background conditions and the baseline environment in which the seeding effects are modeled.

Reply: We appreciate your comment. In the model, the temperature profile is determined by the cloud top temperature and lapse rate. Once the cloud top temperature is given, we can calculate the temperature at different levels. Typically, the lapse rate

in a stratiform cloud is 5-6 K/km, we made sensitivity tests using different lapse rates, and an example is shown in Fig. R1. It is seen that the results are quite similar using a lapse rate of 5 and 6 K/km. In this study, we use 5.5 K/km, and a random temperature perturbation varies between -0.1 K and 0.1 K is applied to each level. Sorry for missing the information about temperature, this has been added to the revised paper.

[Figure]

Figure R1. Vertical profiles of Ze from the simulations with a temperature lapse rate of (a and b) 5 K/km, and (c and d) 6 K/km, and an AgI particle concentration of (a and c) 35 cm$^{-3}$, and (b and d) 350 cm$^{-3}$.

In the model, we assume there is sufficient liquid water or vapor supply, which means once the liquid water is consumed by ice growth, new liquid water can quickly form due to turbulent mixing or updraft (e.g., orographic lifting). Ice formed in a single timestep (1 s) is too low to consume all the liquid water in clouds, it is the continuous ice nucleation that results in complete glaciation if we assume there is no continuous water supply. Therefore, to test the impact of liquid water content on ice growth, it is necessary to consider both the upper limit of LWC and the time duration for ice growth. We note this is related to Comment 2. Please see the more detailed response below.

Clouds in real-world conditions rarely exhibit perfectly adiabatic liquid water content profiles. It is important for the authors to clearly state that the assumption of an adiabatic profile represents a theoretical upper limit of liquid water content available for seeding.

If clouds contain less supercooled liquid water, it is likely that more AgI would be required to achieve comparable seeding effects. To address this, the paper would benefit from a brief sensitivity analysis that explores how variations in background SLW content impact the amount of AgI needed to produce similar reflectivity enhancements. Such an analysis would provide a more realistic comparison and better represent the variability found in natural cloud conditions.

Reply: We appreciate your comment. In the model, we assume there is a continuous water supply. We agree that in real cloud this is not always true, and there is an upper limit of LWC. This would certainly affect the ice growth and the Ze profiles. We made several sensitivity tests in the revised manuscript, including different upper limits of LWC (assuming no continuous liquid water formation), different time durations for ice growth, and different AgI particle concentrations for a limited LWC (Fig. R2a-c). It is seen from the figure that for a model time of 90 minutes, the Ze decreases with decreasing LWC. For a given LWC of 0.2 $gm^{-3}$, ice nucleation and growth in a longer time would consume more liquid water, leading to lower Ze (Fig. 2b), which means the ice formed later on has no sufficient liquid water and vapor to grow. For a given LWC and time duration, more AgI concentration does not mean a larger Ze (Fig. R2c), ice crystals may compete for the limited liquid water and suppress the ice crystal size. In addition, we made sensitivity tests of different turbulent dispersion coefficients and different initial ice particle size distributions (Fig. R2d-f), these are also sources of uncertainties in the model, though the Ze profile is less sensitive to them compared to LWC. This analysis is added in the revised paper, which provides us a better understanding of how the results may vary due to the different environmental conditions.

[Figure]

Figure R2. Vertical profiles of Ze simulation using different (a) upper limits of LWC, (b) time durations for ice growth, (c) AgI particle concentrations with limited LWC, (d) turbulence dispersion coefficients, and (e, d) coefficients in the initial ice particle size distributions.

The study assumes that all clouds examined were initially composed entirely of supercooled liquid water with no background ice crystals. This assumption significantly impacts the interpretation of the seeding effects, as introducing ice crystals in an ice-free environment is likely to produce substantial changes in reflectivity and ice water content. In natural mixed-phase clouds, where ice crystals already exist, the reflectivity changes resulting from seeding may be much more subtle due to the higher baseline of ice content. Therefore, the large reflectivity increases reported in this study could be partially attributed to the idealized initial conditions rather than the inherent efficacy of seeding under more typical, mixed-phase cloud conditions. This raises questions about the generalizability of the findings to real-world scenarios, where the presence of natural ice would likely diminish the detectability of the seeding signature. The authors need to explicitly state this assumption especially in the abstract and conclusions and add a statement at the end on its implications for the interpretation of their results. This is especially true for Section four, where the study attempts to translate the impact of adding additional AgI in a completely supercooled cloud to scenarios with background natural IWC and likely (but unknown) variable supercooled liquid water conditions. The authors need to explicitly state their justification at the beginning of the section

(there is some justification at the end) on why they can make this translation to clouds with natural background ice populations.

Reply: We appreciate your comment. Yes, we agree, in precipitating clouds, the presence of natural ice would likely diminish the detectability of the seeding signature. We do not consider the interaction between the seeded ice and natural ice, and we do not consider the competition for liquid water between the seeded and natural ice. In the paper, as we stated, we assume there is sufficient liquid water and vapor supply, therefore, there is enough liquid water for the growth of both natural ice and seeded ice crystals. Aggregation between natural and seeded ice is expected to further enhance the ice crystal size, which is favorable for seeding signature detection. Therefore, if there is a source for continuous water supply (e.g., orographic lifting), this method is probably valid for precipitating clouds. At least we provide a lower limit of AgI concentration that is needed to detect unambiguous seeding signatures. More AgI may be needed if considering the interaction between natural and seeded ice crystals. We acknowledge that the validation of this model and parametrization in precipitating clouds needs further validation in the future, this discussion is added in the abstract, Section 4, and conclusions.

The paper's conclusions heavily rely on a single comparison with 3D LES data, which raises concerns about the robustness and generalizability of the findings in Sec. 3.2. The 1D simulations appear to involve seeding in a deeper orographic cloud (up to 6 km) over several kilometers, whereas the LES comparison is conducted on a much shallower stratiform cloud, only about 600 m thick, at a single temperature. This discrepancy in cloud depth and structure between the 1D and 3D simulations could significantly influence the modeled seeding effects, particularly in terms of ice growth dynamics, particle fall speeds, and reflectivity changes. Additionally, the thermodynamic environments likely differ between the 1D model runs and the LES simulation, including variations in temperature, moisture profiles, and turbulence intensity, which are critical factors influencing ice nucleation and growth. Without a detailed comparison of these environmental differences, the validity of using the LES as a representative or benchmark for the 1D results is questionable. The paper would benefit from a more thorough examination of how these thermodynamic discrepancies might affect the outcomes to ensure the conclusions are not overly reliant on a single, potentially unrepresentative LES case.

Reply: We appreciate your comment. In the revised paper, we compare the 1D model with the 3D model using 2 cases, a shallow cloud (same as the case in the original manuscript), and a deeper one. The sounding data is originally based on the shallow case (solid line in Fig. R3). To model the deeper case, we modify the temperature and vapor mixing ratio data (dashed line in Fig. R3). The deeper case has a cloud depth of 2 km, seeding is performed at -21 °C. In the shallow case, seeding is performed at -15 °C. The other model configurations are the same as the original manuscript. We acknowledge that this model and the parameterizations need more validation in the future, especially using observational datasets such as Cloudlab (Henneberger et al., 2023).

[Figure]

Figure R3. The initial profiles of (a) temperature and potential temperature, (b) vapor mixing ratio, and (c) U and V components of the wind field. The solid lines indicate the original data used for the shallow cloud, and the dashed lines indicate the modified data used for the deep case. The dark-shaded area (1.3km – 1.9 km) indicates the shallow cloud layer and the light-shaded area (1.3 km - 2.9 km) indicates the deep cloud layer.

Figure R4 shows the modelled Ze, and IWC. It is seen that the Ze and IWC are much greater for the shallow case (Fig. R4a-d) than the deeper one (Fig. R4e-h), because seeding is conducted at -15 °C for the shallow case, while at -21 °C for the deeper case.

[Figure]

Figure R4. Cross sections of Ze (rainbow shading), liquid water mixing ratio (blue shading), and ice mixing ratio (contoured) were obtained from 3D LES simulations for (a-d) the shallow cloud, and (e-h) the deeper case.

Statistically, the 1D model is consistent with the 3D model. Although there are inevitable differences (a few dBZ). Here, we would like to point out there was a mistake (incorrect height data) when plotting the 3D model results in Fig. 6. In the new version, we found the 1D model and 3D model shows similar vertical variation (Fig. R5).

[Figure]

Figure R5. The vertical profiles of Ze simulated using the 1D and 3D models with an AgI particle concentration of (a, c) 35 cm$^{-3}$, and (b, d) 350 cm$^{-3}$ for the (a, b) shallow

and (c, d) deep cases, respectively. The left and right boundaries of the blue shaded area indicate the 95th percentile and the maximum Ze from the 3D model, respectively.

The study attributes reflectivity enhancement primarily to particle growth, but it does not adequately disentangle the contributions from nucleation or increased particle growth versus dispersion effects. A deeper analysis into how much of the observed reflectivity increase due to ice crystal growth, changes in particle size distribution, or enhanced dispersion of particles within the 1D model would provide a clearer understanding of the seeding impacts. This would be nice to show for a single simulation run. Such analysis would clarify the main drivers of reflectivity enhancement and guide more targeted seeding strategies.

Reply: We appreciate your comment. In the revised paper, we made sensitivity tests for different turbulent dispersion coefficients, different coefficients in the ice particle size distribution function, different upper limits of LWC, and different time durations for ice growth. These results are present in Fig. R2, and please see the description for this figure in our reply to comment 2. This figure and related text are added to the revised manuscript.

Specific Comments:

Line 126: Was scavenging of AgI by nucleation accounted for by the 1D model? That one was of the parameterizations in Xue et al. 2013a.

Reply: Yes, we follow Xue et al., 2013a, the fraction of the total AgI particles that are scavenged by cloud droplets is parameterized based on Caro et al. (2024).

Line 126: This would be an appropriate place to discuss seeding strategy, which was limited in the manuscript. Did seeding occur at the specified temperature height? How long did seeding occur? Was it for the entire period of the 1D simulation?

Reply: We appreciate your comment. The seeding strategy is added accordingly. For each run, seeding is performed at a given temperature height, which is randomly selected in the 2500 experiments (1000 in the original paper). Seeding only takes place at the beginning of each run, but ice nucleation by AgI particles keeps occurring. The total AgI particle decreases due to ice nucleation in every time step (1s).

Line 135: Suggest rewording the second half of the sentence: "Radiative cooling at cloud top reduces stability resulting in weak vertical motions at cloud top that may also enhance supersaturation.

Reply: The sentence is reworded to "radiative cooling at cloud top reduces stability resulting in weak vertical motions at cloud top that may also enhance supersaturation".

Line 146: Should this be temperature and supersaturation dependent?

Reply: Yes, it is temperature and supersaturation dependent, this is revised in the paper.

Line 182: Suggest rewording to: terminal fall velocity.

Reply: "terminal velocity" is changed to "terminal fall velocity".

Sec. 2.2: What was the vertical temperature profile used in this analysis? Was it consistent for all 1000 simulation runs?

Reply: We appreciate your comment. In the model, the temperature profile is determined by the cloud top temperature and lapse rate. Once the cloud top temperature is given, we can calculate the temperature at different levels. Typically, the lapse rate in a stratiform cloud is 5-6 K/km, we made sensitivity tests using different lapse rates, and an example is shown in Fig. R1. It is seen that the results are quite similar using a lapse rate of 5 and 6 K/km. In this study, we use 5.5 K/km, and a random temperature perturbation varies between -0.1 K and 0.1 K is applied to each level.

Sec 2.2: A statement needs to be added to this section that the 1D model does not incorporate aggregation or secondary ice production mechanisms, such as rime-splintering or Hallett-Mossop processes, which can significantly alter ice particle size distributions and radar reflectivity. By not including these processes, the model may underestimate the reflectivity in conditions where aggregation is a dominant growth mechanism, or in clouds with abundant SLW or high ice crystal concentrations as a result of SIP.

Reply: We appreciate your comment. The statement is added to the paper: *"The 1D model does not incorporate aggregation or secondary ice production mechanisms, such as the rime-splintering process and shattering of freezing drops, which can significantly alter ice particle size distributions and radar reflectivity. By not including these processes, the model may underestimate the reflectivity in conditions where aggregation*

*is a dominant growth mechanism, or in clouds with abundant SLW or high ice crystal*
*concentrations as a result of SIP.”*

A statement also needs to be added clarifying that nucleated particles that form at a given level are either columnar or plate-like depending on background temperature and remain that way as they descend to the surface.

Reply: We appreciate your comment. The statement is added to the paper: *“Nucleated particles that form at a given level are either columnar or plate-like depending on background temperature and remain that way as they descend to the surface.”*

Sec. 2.4: A statement needs to be added to this section discussing the uncertainties when making radar reflectivity calculations using the assumption of Rayleigh scattering, which is valid for larger particles and longer wavelengths but may not hold at shorter wavelengths (e.g., W-band radar) where reflectivity becomes more sensitive to particle concentration rather than size. This assumption could limit the parameterization's applicability across different radar systems used in cloud seeding operations. A similar statement can be found in the conclusions.

Reply: We appreciate your comment. In fact, we have this discussion in the last section of the paper. According to your suggestion, we add this discussion to Section 2.4.

Fig. 2: Why are there mean layers where particles consistently see decreases in IWC and Ze, if particles are growing with depth? How random is the turbulent kinematic diffusion between the different runs? Are their changes in LWC as a result of variations in the temperature profile under the assumption that the cloud is adiabatic? A figure showing initial conditions would be useful to add at the beginning of the analysis.

Reply: We appreciate your comment. The vertical variation of IWC is because the ice concentration varies with height. Ice crystal concertation is determined by ice nucleation rate, so it is related to turbulence. However, we made a mistake in generating the random numbers of turbulence. This is corrected in the code and the figure is updated (Fig. R6), in the updated figure, we still see a slight variation of IWC, determined by the variation of ice concentration.

[Figure]

Figure R6. Vertical profiles of (a and d) ice concentration, (b and e) IWC, and (c and f) Ze from the simulations with different seeding temperatures and an AgI particle concentration of (a-c) 35 cm$^{-3}$, and (d-f) 350 cm$^{-3}$. The results are obtained based on 10 numerical experiments for each seeding temperature. The shaded area captures the 20th-80th percentile range, and the solid lines are the mean profiles.

Fig 2: A panel showing total number concentration for each distribution would be helpful to the reader to determine how many ice crystals are contributing to IWC and Ze at a given height.

Reply: We appreciate your comment. Ice concentrations are added to the figure (Fig. R6).

Fig. 6: Note in the caption that you are only comparing the -15°C simulation with these LES results in the figure caption.

Reply: We appreciate your comment. The figure is updated and the caption is revised accordingly.

Figure 7: I found Fig. 7 difficult to interpret because there are four parameters. Could it be broken into a three panel plots showing temperature vs AgI, temperature vs height, and then AgI vs height with reflectivity shaded as a function of reflectivity enhancement. That may make it easier to interpret.

Reply: We appreciate your comment. We tried to plot the figure according to your suggestion. However, it seems the impact of cloud depth cannot be well revealed. Therefore, we prefer to use the 3D plot.

[Figure]

Figure R7. Scatter plots of (a) temperature vs AgI, (b) temperature vs depth, and then (c) AgI concentration vs depth with reflectivity shaded as a function of reflectivity enhancement.

Line 272: Suggest removing the sentence starting with "Long-term remote sensing measurements". It seems out of place and doesn't add anything to the discussion without showing additional data.

Reply: We appreciate your comment. This sentence is removed in the manuscript.

Line 433: This assumption needs to be clearly stated at the beginning of this section.

Reply: We appreciate your comment. This assumption is now stated at the beginning of this section, as well as in Section 3.1.

---

## Author Response (AR2)

The reviewer's comments are in black, and responses are in blue.

Dear Editor,

In the revised paper, the color schemes used in Fig. 2 and 3 are modified to allow readers with color vision deficiencies to correctly interpret your findings.

**Reviewer 1:**

Overall, I feel that the manuscript is much approved and the authors addressed all of my comments. I appreciate all of the additional work you put into the paper and feel like it really improved. I look forward to seeing this manuscript published and feel that it will be a unique contribution.

Reply: Again, thank you for the insightful comments, which are helpful to improve the paper.